# WHEN AI CO-SCIENTISTS FAIL: SPOT—A BENCHMARK FOR AUTOMATED VERIFICATION OF SCIENTIFIC RESEARCH

## ABSTRACT

Recent advances in large language models (LLMs) have fueled the vision of automated scientific discovery, often called AI Co-Scientists. To date, prior work casts these systems as generative co-authors responsible for crafting hypotheses, synthesizing code, or drafting manuscripts. In this work, we explore a complementary application: using LLMs as verifiers to automate the **academic verification of scientific manuscripts**. To that end, we introduce SPOT, a dataset of 83 published papers paired with 91 errors significant enough to prompt errata or retraction, cross-validated with actual authors and human annotators. Evaluating state-of-the-art LLMs on SPOT, we find that none surpasses 21.1% recall or 6.1% precision (o3 achieves the best scores, with all others near zero). Furthermore, confidence estimates are uniformly low, and across eight independent runs, models rarely rediscover the same errors, undermining their reliability. Finally, qualitative analysis with domain experts reveals that even the strongest models make mistakes resembling student-level misconceptions derived from misunderstandings. These findings highlight the substantial gap between current LLM capabilities and the requirements for dependable AI-assisted academic verification.

## 1 INTRODUCTION

From simple next-token predictors (Radford et al., 2018; Brown et al., 2020), large language models (LLMs) have evolved to exhibit graduate-level STEM proficiency (Guo et al., 2025a; Rein et al., 2024; Feng et al., 2025), generate hypotheses (Si et al., 2024; Park et al., 2024a), synthesize literature (He et al., 2025), and draft manuscripts (Jain and Jain, 2024). Such advances have driven interest in their deployment as "AI Co-Scientists" (Gottweis et al., 2025; Lu et al., 2024), proving to be viable options in the "generative" role of scientific research. They have rediscovered established findings (Penadés et al., 2025) and generated novel hypotheses worthy of investigation across diverse fields (M. Bran et al., 2024; Pan et al., 2025; DeepMind, 2025). However, despite their widespread usage as "generators" in the forward pass of scientific research, their utility in the backward pass of academic verification or as **verifiers** remains underexplored, a blind spot in which most systems lean on LLM judges (Zheng et al., 2023) without validation on their credibility in reviewing scientific research. Prior research on factual verification has primarily focused on everyday knowledge tasks (Chen et al., 2019; Bekoulis et al., 2021; Zhang et al., 2025), reference-based claim checking (Ortega and Gómez-Pérez, 2025; Kumar et al., 2025), or computer-science disciplines alone (Siegel et al., 2024; Dycke et al., 2022; Baumgärtner et al., 2025). This limits the potential applicability of the proposed benchmarks as evaluation tools for verification systems in AI-driven science research.

In this paper, we introduce SPOT (**S**cientific **P**aper Error Detection), a complex multi-modal academic error verification benchmark, comprising 83 up-to-date manuscripts spanning ten scientific fields with multiple *human-annotated* errors. Given large-scale multi-modal inputs with 12,000 text tokens and 18 images on average, multi-modal LLMs (MLLMs) are tasked with generatively identifying more than one error with varying difficulties in a single paper: *e.g.,*, factual inconsistencies, figure duplications, and mathematical errors. We only select papers published *from* 2024, minimizing the potential contamination with parametric knowledge during evaluation (Bejan et al., 2023). It should be noted that, whereas prior evaluation suites focus on sentence-level fact checks of everyday knowledge (Thorne et al., 2018; Wadden et al., 2020) or on reproducing noisy peer-review feedback (Lin

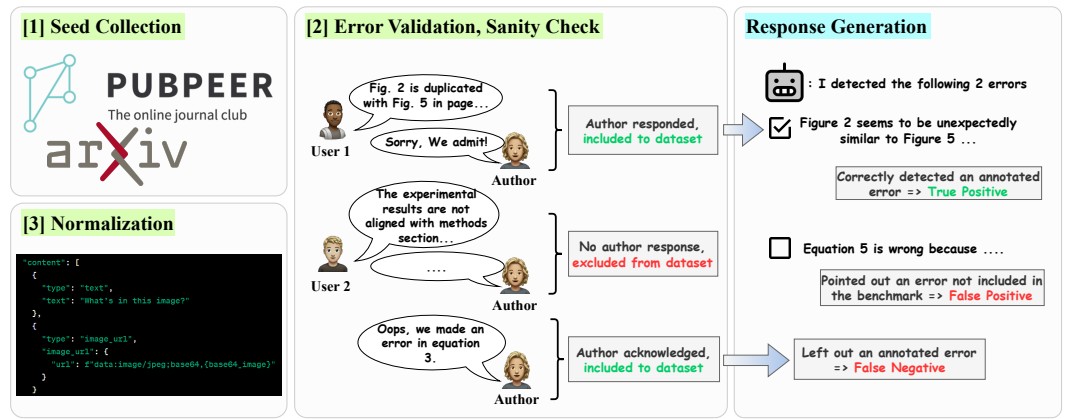

Figure 1: **Overview of SPOT.** Green indicates benchmark construction process, from seed collection through validation to normalization; blue indicates evaluation, where LLM outputs are compared to ground-truth errors and classified as true positives, false positives, or false negatives.

et al., 2023; Shin et al., 2025), SPOT extends verification to the full complexity of frontier-level scientific research. This paper is mainly divided into three parts.

1. **SPOT Benchmark Design Principles** (Section 2): We detail our efforts of multiple automated filtering, author verifications, and human annotations, highlighting our commitment to include confirmed, noncontroversial errors across diverse scientific subdomains.

2. **Model Evaluation and Analysis** (Section 3) We present evaluation results, demonstrating that even the state-of-the-art models struggle on SPOT. Specifically, OpenAI's o3 (OpenAI, 2025a) and Llama-4-Maverick (Meta AI, 2025) achieved 18.4% and 0.9% at pass@1 to name a few. Furthermore, model confidence approaches zero when repeated over eight independent trials, questioning their reliability. We also observe that proprietary reasoning models suffer in detecting figure-related errors, highlighting shortcomings in their multi-modal capabilities. Such results cast serious concerns, revealing significant gap between current AI capabilities and the demands of rigorous scientific verification.

3. **Expert-led Case Studies** (Section 4) We present expert-led case studies in mathematics and materials science, analyzing model outputs to diagnose their failures. Our observations show that models struggle with long-tail knowledge likely absent in web data and extremely long contexts. We also note that, without fully spelled-out derivations, models fail to understand some calculations and overlook domain-specific conventions, making student-like errors.

## 2 SPOT: AUTOMATING ERROR DETECTION IN SCIENTIFIC RESEARCH

In this section, we introduce a detailed overview of SPOT, a complex multi-modal academic verification benchmark with cross-validated scientific manuscripts. We ensure credibility in the error annotations through a cross-validation process between human experts in each field and proprietary language models (Section 2.1). Spanning over ten different fields and six error types (Section 2.2), we introduce evaluation protocols mainly based on precision, recall, and pass@$K$ (Section 2.3).

### 2.1 DATA CURATION

**Stage 1 - Seed Collection** We source our seed manuscripts from two major repositories: (1) WITHDRARXIV (Rao et al., 2024) and (2) PubPeer[1]. First, we extract entries annotated as "factual/methodological/other critical errors" from WITHDRARXIV, a dataset of 14,000 papers and their associated retraction comments. Second, we crawl PubPeer, an anonymous post-publication peer review website, where users flag methodological flaws, image manipulations, and other scientific concerns. Following Ortega (2022), we query initial searches using alphabets, extract high-frequency

---

[1] https://pubpeer.com/

keywords from the returned paper titles, re-query using those keywords, and scrape each paper's metadata (title, authors, venue) alongside the entire comments. We attempted to include medRxiv and bioRxiv, but dropped them due to the low yield (1 and 13 papers each).

**Stage 2 - Automated Filtering**    We apply two GPT-4o (OpenAI et al., 2024)[2] filtering passes. The first retains comment–manuscript pairs that unambiguously pinpoint a specific section, figure, equation, or table, reducing our pool to 1,855 WITHDRARXIV and 25,378 PubPeer samples. The second pass removes reports that require external artifacts (e.g., duplicated images or errors detectable only via external datasets or code). Finally, to avoid overlap with GPT-4o's training cutoff , we filter for papers published after 2024, yielding 58 WITHDRARXIV and 215 PubPeer samples.

**Stage 3 - Error Validation by Original Authors**    For remaining manuscripts, we only retain those the *original authors directly confirmed*. Specifically, we only retain PubPeer comments followed by an explicit author response acknowledging the mistake and treat WITHDRARXIV self-retractions as definitive evidence of a critical error. In all cases where the author themselves admits the problem, we take this acknowledgment as confirmation of a genuine error. While some errors may appear to be evident, we do not include any error with explicit acknowledgment from the original authors, as many of the work cover ungoing areas of research, which remain unsettled in the scientific discourse.

**Stage 4 - Sanity Check from Human Annotators**    We further apply a two-stage human validation with mutually exclusive annotators. First, with part of the authors as human annotators, we manually validate if remaining flagged issues fulfill three conditions: (1) self-contained, (2) identifiable, and (3) explicitly acknowledged by the original authors. For those which satisfy the conditions, We retrieve the archived PDF to verify that the error remains visible, then document a concise description of the problem, quote the author's acknowledgement verbatim, and assign both an error category and a severity rating—proxied by the form of the author's response (erratum versus retraction). Afterwards, the second group conducted a comprehensive audit to ensure consistent application of these standards. The final SPOT benchmark comprises 83 manuscripts with 91 annotated errors. Although modest in size, our dataset aligns with recent trends toward compact, high-quality benchmarks: MT-Bench (80 items) (Zheng et al., 2023), GPQA-D (198 items) (Rein et al., 2024), AIME 2024/2025 (30 items each) (MAA, 2024), and USAMO 2025 (6 items) (Petrov et al., 2025).

**Stage 5 - Normalization**    We normalize manuscripts in PDF format into text and image sets. While prior benchmarks in manuscript error detection (Baumgärtner et al., 2025) and AI-assisted science (Seo et al., 2025) have relied on raw PDFs or text-only inputs, this approach offloads document understanding to OCR and parsing modules rather than the LLM itself, thereby conflating upstream parser failures with downstream model errors. Instead, we process all the documents for usage. We first employ Llama-Parse[3] to convert each PDF into Markdown and capture high-fidelity screenshots of every figure, table, and equation. In pilot experiments, OCR failures, particularly in mathematical expressions, led downstream models to misinterpret formatting artifacts as errors. To address this, we introduce a refinement stage. For each page, the initial OCR text and screenshots (one full-page image plus isolated equations and paragraphs, roughly eight images per page) are sent to GPT-4.1 for correction. Finally, we conduct a manual audit of all processed pages to ensure that every flagged error remains visible and accurately represented in the OCR output.

## 2.2 BENCHMARK STATISTICS

**Error Types**    We derive the six categories in Table 1 inductively from our annotations rather than setting a priori. As we review each error, we group similar cases. This is to capture the true distribution of errors existing in manuscripts. During this process, figure-duplication instances initially overwhelmed the dataset, so we filtered based on severity and paper category to prevent a single type from dominating.

**Paper Subjects**    We present general statistics in Table 1. We classify each paper into ten research domains: Mathematics, Physics, Biology, Chemistry, Materials Science, Medicine, Environmental

---

[2]Using version gpt-4o-2024-08-06

[3]https://www.llamaindex.ai/llamaparse

Table 1: **Overview of SPOT.** *Left*: High-level statistics—83 manuscripts, 91 errors from 47 paper sources; tokens per manuscript (mean ± std., range) and images per manuscript (mean ± std., range). All token counts were computed using the GPT-4o (Hurst et al., 2024) tokenizer from `tiktoken` (OpenAI, 2025b). *Right*: Six error categories with concise descriptions and instance counts in parentheses.

| Benchmark Statistics | | Category | Descriptions |
|---|---|---|---|
| *General* | | Equation / Proof (37) | Incorrect mathematical derivations |
|     Total Manuscripts: 83 | | | |
|     Total Errors: 91 | | Figure Duplication (27) | Reused or manipulated images |
|     Total Paper Sources: 47 | | | |
| *Tokens* | | Data Inconsistency (18) | Mismatched values between text, tables, and figures |
|     Avg (std): $12,887_{7,421}$ | | | |
|     Max / Min: $46,441/1,207$ | | Statistical Reporting (4) | Misused statistical values or inappropriate tests |
| *Images* | | | |
|     Avg (std): $17.5_{20.1}$ | | Reagent Identity (3) | Mislabeled or incorrect materials |
|     Max / Min : $80/0$ | | | |
| *Error severity* | | Experiment Setup (2) | Missing controls or misreported protocols |
|     Errata / Retract: 59/32 | | | |

Science, Engineering, Computer Science, and Multidisciplinary, based on its journal venue or arXiv subject. In Figure 2, we observe clear domain patterns: mathematics, computer science, and physics papers skew toward equation/proof flaws; biology toward figure-duplication. 76 manuscripts out of 83 contain a single error, six contain two, and one paper has the maximum of three annotated errors. We proxy error severity by the authors' post-publication response: 59 errors were addressed via errata, while 32 led to full retractions. Retractions are concentrated mostly in equation/proof cases. Manuscripts span 1k–46k tokens and include 0–80 figures, creating a long context, multimodal, and figure-rich benchmark far exceeding the scale and complexity of existing error-detection datasets. Although longer papers tend to include more figures, the relationship is weak (Pearson's $r = 0.19$), highlighting diverse presentation styles across fields.

## 2.3 EVALUATION PROTOCOL

We provide the full paper as interleaved text and image data, followed by the prompt to return every error with each error's location (section, figure, equation, or table), accompanied by a description. The output is prompted to be a structured JSON format (see Appendix F for an example).

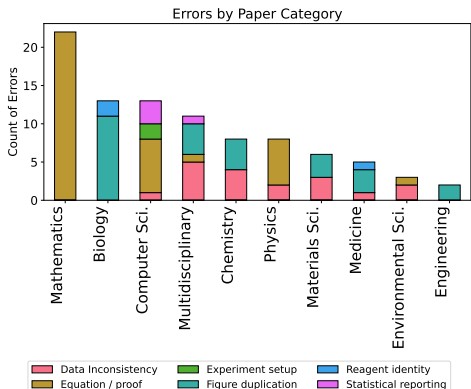

Figure 2: **Distribution of annotated errors by research domain and error type.**

**Evaluation Metric** We mainly evaluate verification performance through precision, recall, and pass@$K$. A predicted error is counted as a true positive (TP) only when the model's reported location matches a benchmark annotation and an LLM confirms they indicate the same error[4]. All others, including those at non-annotated locations or those matching an annotated location but with a different description, are considered false positives (FP), and any benchmark annotation the model fails to predict is a false negative (FN). We treat the error annotations included in SPOT as exhaustive: any model-reported error not matching an annotation is counted as a false positive. Although models could, in principle, flag genuine errors outside our annotations, through case studies later in this paper, we notice such cases are highly unlikely. To summarize model performance, we report Precision and Recall:

$$\text{Precision} = \frac{\text{TP}}{\text{TP} + \text{FP}}, \qquad \text{Recall} = \frac{\text{TP}}{\text{TP} + \text{FN}}. \qquad (1)$$

---

[4]GPT-4.1 is used to compare predicted error descriptions against benchmark annotations as a similarity check (Ni et al., 2024); the LLM does not evaluate the errors' correctness or severity.

Table 2: **Performance of ten models on the SPOT dataset.** The Think column denotes the use of test-time scaling. Precision, Recall, pass@1 and pass@4 (all in %) are reported as mean and standard deviation (in parentheses) over eight independent trials. The highest value in each column is **bolded**, and the second-highest is underlined. Detailed evaluation results are available in Appendix G.

| Models | Think | Precision (%) | Recall (%) | pass@1 (%) | pass@4 (%) |
|--------|-------|---------------|------------|------------|------------|
| o3 (2025-04-16) | ✓ | $\mathbf{6.1_{1.3}}$ | $\mathbf{21.1_{4.4}}$ | $\mathbf{18.4_{2.1}}$ | $\mathbf{37.8_{1.8}}$ |
| GPT-4.1 (2025-04-14) | ✗ | $2.8_{0.8}$ | $6.0_{1.6}$ | $6.6_{1.7}$ | $17.8_{1.5}$ |
| Gemini-2.5-Pro (preview-03-25) | ✓ | $\underline{3.1_{1.7}}$ | $\underline{10.1_{5.6}}$ | $7.8_{3.8}$ | $\underline{25.9_{4.0}}$ |
| Gemini-2.0-Flash-Lite (001) | ✗ | $1.0_{0.8}$ | $1.6_{1.3}$ | $1.5_{1.0}$ | $6.0_{1.5}$ |
| Claude-3.7-Sonnet (20250219:Think) | ✓ | $3.0_{1.3}$ | $6.0_{2.4}$ | $5.5_{1.7}$ | $18.6_{2.8}$ |
| Claude-3.7-Sonnet (20250219) | ✗ | $3.2_{1.5}$ | $5.8_{2.7}$ | $4.5_{1.9}$ | $14.1_{1.6}$ |
| Qwen2.5-VL-72B-Instruct | ✗ | $0.6_{1.2}$ | $0.4_{0.7}$ | $0.4_{0.6}$ | $1.7_{1.0}$ |
| Qwen2.5-VL-32B-Instruct | ✗ | $1.9_{2.0}$ | $1.9_{1.7}$ | $2.0_{1.5}$ | $5.6_{1.6}$ |
| Llama-4-Maverick | ✗ | $2.0_{2.6}$ | $0.9_{1.2}$ | $0.9_{1.0}$ | $3.3_{1.2}$ |
| Llama-4-Scout | ✗ | $0.8_{1.0}$ | $1.9_{2.3}$ | $1.8_{2.0}$ | $7.2_{3.1}$ |

*Precision* quantifies the proportion of the model's flagged errors that match benchmark annotations, penalizing unexpected predictions, and is most appropriate when false positives impose significant review overhead or undermine confidence in the tool's outputs. *Recall* quantifies the proportion of annotated errors the model successfully identifies, penalizing missed detections. In practice, users concerned about model hallucinations or the impact of unannotated flags should focus on Precision. In contrast, those seeking comprehensive error coverage, or who doubt the exhaustiveness of our annotations, should emphasize Recall.

Following Kulal et al. (2019) and Chen et al. (2021), to capture how error detection improves with multiple attempts, for $K$ runs per paper, we define :

$$\text{pass@}K = \frac{1}{\sum_{i=1}^{N} |G_i|} \sum_{i=1}^{N} \sum_{g \in G_i} \mathbf{1}\Big[\exists\, s \in \{1, \dots, K\} : g \in p_i[s]\Big], \tag{2}$$

where $G_i$ is the set of annotated errors in paper $i$ and $p_i[s]$ the set predicted in the $s$-th run. With 83 papers and 91 total errors we generate $N = 8$ independent runs per paper. For each pass@$K$ we draw $K$ runs without replacement from the eight, repeat this resampling $B = 1000$ times, and report the mean and standard deviation of the resulting bootstrap distribution for $K \in \{1, 4\}$.

## 3 MAIN RESULTS AND ANALYSIS

In the following sections, we evaluate six proprietary models: OpenAI o3 (OpenAI, 2025a), GPT-4.1 (OpenAI, 2025c), Google Gemini 2.5 Pro (Google Cloud, 2025a), Gemini 2.0 Flash Lite (Google Cloud, 2025b), Anthropic Claude 3.7 Sonnet:Thinking (Anthropic, 2025), and Claude 3.7 Sonnet and four open models: Qwen 2.5-VL-72B/32B-Instruct (Bai et al., 2025), and Llama 4 Maverick/Scout (Meta AI, 2025). We select the most capable models per family and observe that these models already score near zero in SPOT. Accordingly, as smaller models are unlikely to perform any better, we do not include them in our evaluations. All models are accessed via APIs, and each call is retried up to three times; those that still fail or are cut off due to length limits are marked incorrect.

### 3.1 MAIN RESULTS

Table 2 compares ten multi-modal LLMs on SPOT. o3 achieves the highest scores, with $6.1\% \pm 1.3$ precision, $21.1\% \pm 4.4$ recall, and a 37.8% pass@4. It is followed by Gemini-2.5-Pro (3.1%, 10.1%, 25.9%), Claude-3.7-Sonnet:Thinking (3.0%, 6.0%, 18.6%), and GPT-4.1 (2.8%, 6.0%, 17.8%). The lighter proprietary variants, Gemini-2.0-Flash-Lite and the non-Thinking Claude-3.7-Sonnet, score marginally above zero. Surprisingly, open-source models such as Qwen2.5-VL-72B-Instruct and Llama-4-Maverick, which match proprietary models on existing multi-modal benchmarks like MMMU (Yue et al., 2024) or MathVista (Lu et al., 2023), perform far worse on SPOT. As shown in Figure 3, (1) the performance gap between o3 and Llama-4-Maverick is widest on SPOT (ours)

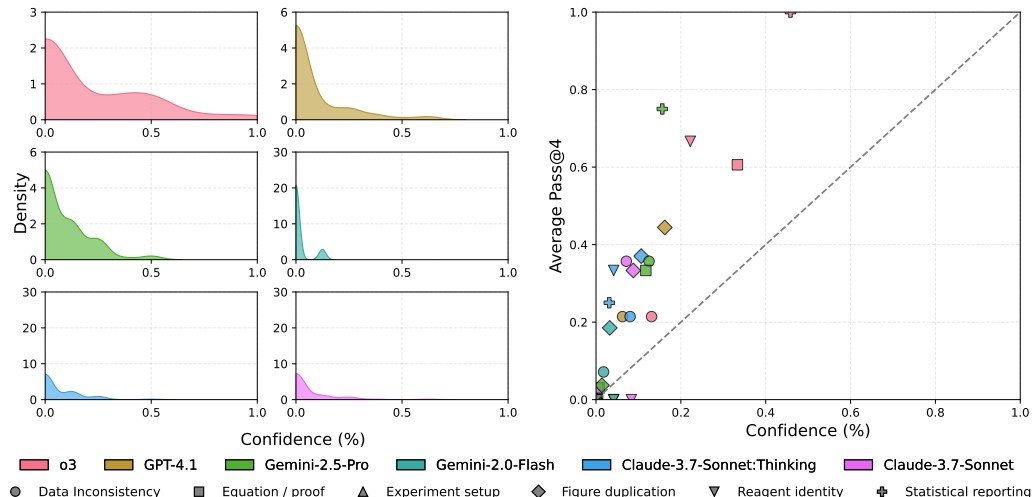

Figure 4: **Category-specific performance and calibration of six LLMs on SPOT.** *Left*: Kernel density estimates of each model's reported confidence; all six models predominantly express very low confidence. *Right*: Scatter plot of mean reported confidence (see Appendix C for further details) versus pass@4 for each model (color), broken down by error type (shape). The dashed diagonal marks perfect calibration.

($\Delta = 20.2$ pp), and (2) SPOT is the only benchmark where Llama-4-Maverick's score collapses to near zero (0.9 %). While neither proprietary nor open-source models fully satisfy the requirements of practical deployments of error-detecting AI systems, open-source models lag far behind in domain-specific rigor and robust error-detection capabilities essential for scientific applications.

**A New Challenging Benchmark for STEM.** Figure 3 illustrates the performance of o3 on six benchmarks: MathVista (Lu et al., 2023), MMLU-Pro (Wang et al., 2024), GPQA Diamond (Rein et al., 2024), MMMU (Yue et al., 2024), HLE (Phan et al., 2025) and SPOT (recall). o3 exceeds 80% on the first four benchmarks, demonstrating robust general reasoning and code understanding. However, performance drops to roughly 20% on HLE, a curated set of frontier, research-level academic questions, and remains similarly low on SPOT (21.1%). This drop in performance underscores the difficulty of spotting errors in lengthy scientific text and figures.

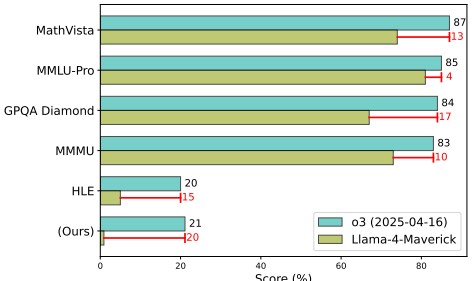

Figure 3: **Performance of o3 and Llama-4-Maverick across six challenging STEM benchmarks.** The short red horizontal lines mark the gap $\Delta = $ o3 $-$ Llama-4-Maverick for each benchmark.

**Reasoning Models Excel at Equations but Falter on Figures** The right panel of Figure 4 presents the performance of six models across each category. In the Equation/Proof category, o3 leads with a 62.6% (pass@4), followed by Gemini-2.5-Pro at 36.4%, while all other models remain below 5%, underscoring o3's superior mathematical reasoning. Surprisingly, GPT-4.1 achieves a 44.4% in the Figure Duplication category, outperforming Claude-3.7-Sonnet Thinking (33.3%), o3 (0%), and Gemini-2.5-Pro (0%), revealing a weakness in figure analysis in reasoning models.

### 3.2 UNRELIABILITY OF MISCALIBRATED MODELS.

Alongside pass@4, calibration (Guo et al., 2017; Ovadia et al., 2019) indicates how much we should trust a model's predictions. In error detection, where false positives can incur substantial time and labor, knowing when to trust a model is crucial. For each error category, we assess calibration by comparing the model's actual performance, measured as its average pass@4 rate, with its self-estimated confidence. For details on how the confidence is derived see Appendix C.

However, Figure 4 (right) shows that confidence correlates only weakly with pass@4, and the left panel reveals that most models report very low confidence, clustering near zero. Across 498

Table 3: **Multi-modality ablation for 13 models**: recall and pass@4 (in %) are reported as mean (std) over eight independent trials. The left panel shows each model's performance with multi-modal inputs; the right panel shows performance on the text-only subset of SPOT (48 figure-independent instances), including additional unimodal LLMs (DeepSeek-R1, DeepSeek-V3, Qwen3-235B-A22B). The highest value in each column is **bolded**, and the second-highest is underlined. Detailed evaluation results are available in Appendix G.

| Models | Think | Multi-Modal | | Text-Only | |
|---|---|---|---|---|---|
| | | Recall (%) | pass@4 (%) | Recall (%) | pass@4 (%) |
| o3 (2025-04-16) | ✓ | $34.6_{7.1}$ | $61.1_{2.9}$ | $25.7_{7.1}$ | $56.2_{4.2}$ |
| GPT-4.1 (2025-04-14) | ✗ | $0.5_{0.9}$ | $2.0_{1.4}$ | $8.4_{2.5}$ | $19.8_{2.7}$ |
| Gemini-2.5-Pro (preview-03-25) | ✓ | $13.7_{8.6}$ | $34.8_{6.1}$ | $6.9_{3.1}$ | $17.0_{2.8}$ |
| Gemini-2.0-Flash-Lite (001) | ✗ | $0.4_{0.9}$ | $2.1_{1.3}$ | $1.9_{1.4}$ | $8.0_{1.8}$ |
| Claude-3.7-Sonnet (20250219:Think) | ✓ | $2.9_{2.3}$ | $8.5_{1.7}$ | $5.0_{2.3}$ | $17.0_{3.1}$ |
| Claude-3.7-Sonnet (20250219) | ✗ | $1.9_{2.1}$ | $4.8_{1.5}$ | $5.8_{3.1}$ | $15.2_{2.8}$ |
| DeepSeek-R1 | ✓ | – | – | $14.8_{3.8}$ | $38.6_{3.3}$ |
| DeepSeek-V3 (0324) | ✗ | – | – | $1.9_{1.1}$ | $6.7_{2.1}$ |
| Qwen3-235B-A22B | ✓ | – | – | $15.4_{6.2}$ | $38.2_{3.1}$ |
| Qwen2.5-VL-72B-Instruct | ✗ | $0.0_{0.0}$ | $0.0_{0.0}$ | $4.7_{2.2}$ | $11.2_{2.5}$ |
| Qwen2.5-VL-32B-Instruct | ✗ | $0.4_{0.8}$ | $1.7_{0.9}$ | $1.1_{1.5}$ | $3.0_{1.2}$ |
| Llama-4-Maverick | ✗ | $0.5_{0.9}$ | $2.1_{1.4}$ | $0.8_{1.0}$ | $3.5_{1.5}$ |
| Llama-4-Scout | ✗ | $0.4_{0.8}$ | $2.0_{1.4}$ | $1.6_{2.1}$ | $5.9_{2.5}$ |

model–instance evaluations (83 instances × six models), we observe only two cases (both from o3) of full confidence, highlighting the widespread difficulty of reliably detecting errors in scientific manuscripts. These findings demonstrate substantial variability across categories and reaffirm that current LLMs remain unreliable for scientific error detection.

### 3.3 IMPACT OF MULTI-MODALITY IN DETECTING SCIENTIFIC ERRORS

To isolate the impact of images, we create a text-only subset by removing all instances from the figure-duplication and any data-inconsistency category that necessitate figures. This yields 48 instances in which errors can be detected using text alone. Table 3 compares model performance on the selected instances under multimodal and text-only conditions. The left panel reports each model's accuracy on these 48 cases with figures included; the right shows performance after stripping out all figures. In the text-only setting, we add three unimodal LLMs: DeepSeek-R1 (Guo et al., 2025b), DeepSeek-V3 (Liu et al., 2024), and Qwen3-235B-A22B (Yang et al., 2025).

We observe two key findings. First, most models improve in recall and pass@4 when removing images, suggesting that figures usually act as distractors. The exceptions are o3 and Gemini-2.5-Pro, which see a modest drop without visual inputs. This indicates that they have been leveraging figures to understand the paper rather than treating them as mere auxiliary signals. Second, the divide between proprietary and open models is vast in the multi-modal setting, proprietary systems maintain substantial recall (e.g., o3 at 34.6 %, Gemini-2.5-Pro at 13.7 %) and pass@4, whereas open-source models collapse to near zero.

## 4 CASE STUDIES : EXPERT-LED REVIEW

To analyze model output in detail, we select two withdrawn manuscripts, each from mathematics and materials science, for a qualitative review. A domain expert evaluated a paper and model outputs, either a researcher with relevant publications or a PhD-trained postdoc in the field. Reviewers are provided the LLM-flagged "errors" from o3 and Gemini 2.5 Pro alongside the official withdrawal notices. They are asked to verify whether the model has missed any benchmarked errors. Moreover, they are required to assess each flagged issue that falls outside our annotations to determine if any presumed false positives correspond to valid flaws. We consulted the original authors to verify the disputed issues whenever a reviewer remained uncertain.[5]

---

[5]Due to space constraints, we show only excerpts of model responses and one case study per domain; for the complete results, see Appendix D.

## 4.1 Mathematics : Petersen and Tommas(2024)

Petersen and Tommasi (2024) studies the configuration spaces of points in algebraic varieties with a multiplicative decomposition, and discusses some applications such as the cohomology of moduli stacks of hyperelliptic curves. It was withdrawn because of a gap that lies in the core arguments of Theorem 1.8 and Theorem 1.13 which invalidates the bulk of the paper.

Both o3 and Gemini-2.5-Pro exclusively flag issues in Section 3. Ironically, this is the only part of the manuscript **not** affected by the actual mathematical gap. o3 criticizes the calculation of $H^k(M_{1,1}, V_\ell)$ in Section 3.3, claiming that the use of Eichler-Shimura isomorphism is wrong without giving too much details. But there is no mistake in this part and the claim is hallucinated. It is worth noting that applying Eichler-Shimura isomorphism in a similar context as the paper is exceedingly rare in internet data. The formula may also appear differently than how the Eichler-Shimura isomorphism is presented in common internet sources, whose equivalence requires genuine understanding of multiple subjects, such as variation of Hodge structures and the theory of modular forms.

> Eichler–Shimura is quoted as $H^1(M_{1,1}, V_\ell) \cong S_{\ell+2} \oplus \mathbb{Q}(-1-\ell)$, but the Tate (Eisenstein) summand should be $\mathbb{Q}(-1)$, independent of $\ell$. The extra "$-\ell$" is a mis-twist.

Figure 5: o3's feedback on Petersen and Tommasi (2024).

On the other hand, Gemini-2.5-Pro tackles Theorem 3.8 as shown in Figure 6. Despite doing reasonable calculations, the model makes a false claim on $H^0(A_{1,1}, V(2,2))(-1)$, resembling mistakes students could make after recollecting a wrong fact.

> The stated result $H^1(M_2, V(2,2)) = \mathbb{Q}(-3)$ contradicts the Gysin-sequence derivation (Sec. 3.1), which yields $H^1(A_2, V(2,2)) = 0$, together with $H^0(A_{1,1}, V(2,2))(-1) = \mathbb{Q}(-3)$ (Sec. 3.5) and the non-vanishing/isomorphism of the Gysin map $G : H^0(A_{1,1}, V(2,2))(-1) \to H^2(A_2)$, thereby implying $H^1(M_2, V(2,2)) = 0$.

Figure 6: Gemini-2.5-Pro's feedback on Petersen and Tommasi (2024).

## 4.2 Material science : Ye et al. (2025)

In our materials science case study, we select Ye et al. (2025), which uses TfOH to synthesize covalent organic frameworks (COFs). However, it mislabels several samples in its figures. These errors are easier to spot than those in the mathematics papers and would be obvious to any attentive reader.

> Optimal TfOH concentration is stated as 0.006–0.016 M, yet all optimisation and standard syntheses use 0.2 M TfOH. Internal inconsistency invalidates the universality of the claimed optimum.

Figure 7: o3's feedback on Ye et al. (2025).

Errors pointed out by o3 exemplify failures in both long-range context comprehension and multi-hop reasoning. For instance, in Figure 7, o3 does not realize that the optimal concentration value reported by the authors (0.006–0.016 M) is the concentration of the final mixture, while the the second value (0.2M) is the concentration of the acid before being added to the final mixture. This misunderstanding likely arises because the optimal concentration in the final mixture is mentioned only once, and the explicit calculation is not shown throughout the manuscript. As a result, o3, having seen references only to the concentration before mixture, fails to infer the relationship between the two values.

In (A) of Figure 8, Gemini 2.5 Pro seems to make a "reading" mistake, attributing the second facet pair to TAPPy-TFPPy-COF when it in fact describes TAPPy-BPTC-COF. Notably, however, in (B), it

(A) There is a contradiction in the indexing of PXRD peaks for TAPPy-TFPPy-COF (Figure 6H). The peaks are initially assigned to facets including (020): 'TAPPy-TFPPy-COF displayed peaks [...]

(B) The BET surface area for the scaled-up TFPPy-PDA-COF is reported as '1606 $cm^2\,g^{-1}$'. The correct unit is '$m^2\,g^{-1}$.' This unit error misrepresents the surface area by a factor of 10,000, constituting a fundamental data-reporting mistake.

Figure 8: Two of Gemini 2.5 Pro's feedback on Ye et al. (2025).

notices a potential error in the units, where a certain compound was assigned a surface area 10000x smaller than all the other compounds in the same family. Because the authors do not mention this extreme property of this material, we suspect that this is a real typo. While not severe, **this error is the only instance** in which we observe an LLM identifying an unannotated but genuine error.

In summary, case studies demonstrate that current LLM models struggle in SPOT. In mathematics, both models mistakenly flagged issues unrelated to the genuine critical error, revealing difficulties in handling rare and complex mathematical formulations. In contrast, in materials science, while o3 misunderstood the experimental details due to poor contextual reasoning, Gemini 2.5 Pro successfully identified an actual unannotated error involving units.

## 5 RELATED WORKS

**AI Co-Scientists**   Recent breakthroughs have pushed LLMs to PhD-level performance on STEM benchmarks (Rein et al., 2024), driving them as generators of the scientific forward pass, encompassing hypothesis generation (Si et al., 2024), experimental planning (Seo et al., 2025), and manuscript drafting (Jain and Jain, 2024). Such systems, or AI Co-Scientists (Lu et al., 2024; DeepMind, 2025), employ agent-based pipelines that mirror the stages of scientific research. However, concurrent works often omit a rigorous backward pass or "verification" and instead rely on LLM judges (Zheng et al., 2023). Yet, prior studies demonstrate that LLM judges may fail on complex tasks (Son et al., 2024), allowing factual and methodological errors to remain undetected. This compromises the reliability of AI-driven research. Notably, verifiability has long been central to scaling AI progress: self-supervised learning employs next-token prediction as a provable training objective (Jernite et al., 2017); and reinforcement learning uses verifiable rewards (Guo et al., 2025b) for alignment. Likewise, we posit that robust scientific verification must underpin reliable LLM-driven scientific research.

**Automating Scientific Verification**   Two research strands, fact verification and automated peer review generation, may appear related to SPOT, but each has critical limitations. Prior fact verification benchmarks (Thorne et al., 2018; Wadden et al., 2020) concentrate on claims at the sentence level and rely on text inputs to assess consistency with references. Automated peer review systems draw almost entirely on computer science publications (Gao et al., 2024; Baumgärtner et al., 2025), restricting their disciplinary coverage. These approaches measure success by matching past reviews via metrics such as ROUGE (Zeng et al., 2024) rather than detecting errors. They also overlook the inherent noise in peer review reports (Cortes and Lawrence, 2021; Bonavia and Marin-Garcia, 2023) and seldom apply adequate quality control or validate ground truth. Our work sets apart, by using expert and automated validation to distill only genuine mistakes into SPOT. Additionally, we package full, multimodal papers into models at inference, mirroring real-world academic verifications.

## 6 CONCLUSION

In this paper, we introduce SPOT, a multimodal error-detection benchmark that captures the full complexity of frontier-level scientific research. Each instance averages 12,000 text tokens and 18 images, posing a significant challenge for current large language models: OpenAI's o3 and Google's Gemini 2.5 Pro achieve pass@1 scores of only 18.4 % and 7.3 %, respectively. Our expert-led case studies further show that these models fall short in long-tail domain knowledge and implicit multi-step calculations. Together with the rise of interest in AI Co-Scientists, these results highlight the need for further research in robust verification systems to ensure reliability in AI-driven research workflows.

## ETHICS STATEMENT

We used ChatGPT to refine the writing and assist with coding. Before release, we removed copyrighted material from the dataset in accordance to publisher policies.

## REPRODUCIBILITY STATEMENT

Evaluation prompts and processing details are provided in Appendix F. We release the code, data-generation scripts, training configurations, and evaluation pipelines at `https://anonymous.4open.science/r/SPOT-anon-C0F7/`. The dataset will be released on Hugging Face.

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

## A  LIMITATIONS

**Benchmark Coverage**   By prioritizing copyright compliance(Appendix E), contamination prevention, and annotation accuracy, SPOT remains relatively modest in size. We leave the expansion of this effort to create larger, more diverse benchmarks that span additional scientific disciplines and error categories to future works.

**Annotation Validity and Evaluation Protocol**   All errors in SPOT are verified by explicit author acknowledgments or retraction notices, but the complexity of scientific manuscripts means some true errors may be unannotated. Our case studies reveal that false negatives can arise from the following cases:

1. The author's note contains an error location that does not sufficiently cover all the affected results.

2. There exist smaller errors unrelated to the main technical error in the preprint.

Conversely, false positives may occur when:

1. An LLM correctly points out a theorem that contains an error, but the content in the LLM's response is still irrelevant.

We therefore recommend a secondary expert review, particularly for domains with complex logical dependencies or deep specialization, to validate and refine model-flagged errors.

## B  ADDITIONAL ANALYSIS

### B.1  IMPACT OF CONTEXT LENGTH IN DETECTING SCIENTIFIC ERRORS

In Table 2, we evaluate each model on complete manuscripts, which can span up to 140,000 characters and 90 figures. However, LLMs still struggle to synthesize long-context information (Vodrahalli et al., 2024); to isolate the effect of context length on error detection, we extract the page containing the ground-truth error and rerun the same detection prompt on this shorter segment. Comparing the `full-paper` and `segment-only` settings decouples long-context processing from core error-detection ability. For this ablation, we conduct experiments on a subset of 36 instances, excluding the Equation/Proof category—mathematical papers often rely on global notation and prior results, making single sections insufficient—and we omit any errors that span multiple sections.

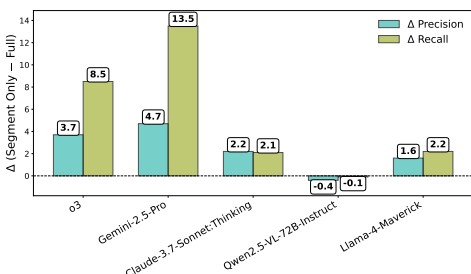

Figure 9: **Impact of context length on error detection.** Each bar shows Δ = (segment-only - full-paper) for precision and recall across five models (o3, Gemini 2.5 Pro, Claude 3.7-Sonnet:Thinking, Qwen 2.5-VL-72B-Instruct, Llama-4-Maverick).

Figure 9 plots Δ = (segment-only – full-paper) for precision and recall. Gemini-2.5-Pro leads with gains of +4.7 precision and +13.5 recall. o3 follows at +3.7/+8.5, then Claude-3.7-Sonnet at +2.2/+2.1, and Llama-4-Maverick at +1.6/+2.2—showing that long-context processing often masks their true error-detection performance. o3's smaller gains reflect the removal of Equation/Proof cases, its original strength. Qwen2.5-VL-72B-Instruct shows almost no change (–0.4 precision, –0.1 recall), indicating a fundamental limit in its error-detection capability rather than a context-length issue.

### B.2  IMPACT OF TEST-TIME SCALING IN DETECTING SCIENTIFIC ERRORS

Test-time scaling involves adjusting the inference budget (Jones, 2021), such as the depth of reasoning or number of solution paths explored (Son et al., 2025), to boost model performance on complex tasks. This approach is widely adopted in STEM and reasoning benchmarks (Snell et al., 2024), where allocating more computational effort to inference has been shown to yield higher performance.

We use OpenAI's o4-mini series (OpenAI, 2025a) for our experiments and vary the "reasoning effort" parameter across low, medium, and high settings.[6]

In Figure 10, we see that o4-mini's error-detection performance increases almost linearly with higher reasoning effort, demonstrating that scaling computation at test time effectively boosts accuracy. This finding is consistent with how specialized "Thinking" modes (e.g., Claude 3.7-Sonnet:Thinking VS-Claude 3.7-Sonnet) and reasoning-trained models (DeepSeek-R1 vs. DeepSeek-V3) deliver similar boosts, also in line with recent error-detection literature (Ahuja et al., 2025).

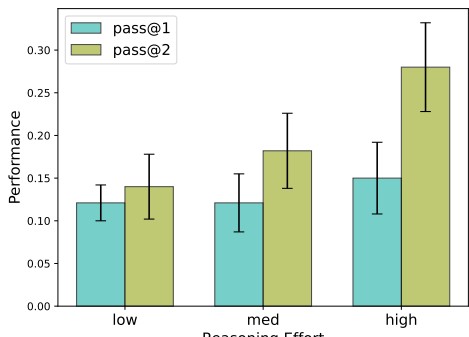

Figure 10: **Performance of o4-mini with varying reasoning effort.** Performance is reported from three independent trials.

## C   ESTIMATING CONFIDENCE FOR pass@$K$

Given $N$ papers with ground-truth error sets $G_1, \ldots, G_N$, we define the pass@$K$ metric as

$$\text{pass@}K = \frac{1}{\sum_{i=1}^{N} |G_i|} \sum_{i=1}^{N} \sum_{g \in G_i} \mathbf{1}\Big[\exists\, s \in \{1, \ldots, K\} : \ g \in p_i[s]\Big], \tag{3}$$

where $p_i[s]$ denotes the set of errors predicted in the $s$th run for paper $i$. This captures the fraction of all ground-truth errors detected in at least one of $K$ independent attempts.

### C.1   UNBIASED PER-ERROR CONFIDENCE

To assign each ground-truth error $g \in G_i$ a *confidence* score, we perform $n$ independent runs (here $n = 8$) and let $c_{i,g}$ be the number of runs in which $g$ is detected. The probability that all $K$ fresh attempts miss $g$ is

$$\frac{\binom{n - c_{i,g}}{K}}{\binom{n}{K}}, \tag{4}$$

So one minus this quantity is the probability of $\geq 1$ success (Chen et al., 2021). Hence the unbiased estimator for the pass@$K$ probability of error $g$ is

$$\hat{p}_{i,g} = 1 \ - \ \frac{\binom{n - c_{i,g}}{K}}{\binom{n}{K}}. \tag{5}$$

### C.2   AGGREGATING CONFIDENCE AND CALIBRATION

We then aggregate these per-error confidences into an overall self-estimated confidence:

$$\text{Confidence} = \frac{1}{\sum_{i=1}^{N} |G_i|} \sum_{i=1}^{N} \sum_{g \in G_i} \hat{p}_{i,g}. \tag{6}$$

---

[6]While recent work has demonstrated similar budget controlling strategies for open models (Muennighoff et al., 2025), the full-size MLLMs (Llama-4-Maverick totaling 402B parameters) were too large to host for multi-thousand-token generations.

# D  CASE STUDY

## D.1  MATHEMATICS

We selected Petersen and Tommasi (2024) and Park et al. (2024b), both in the area of algebraic geometry in pure mathematics. Petersen and Tommasi (2024) studies the configuration spaces of points in algebraic varieties with a multiplicative decomposition, and discusses some applications such as the cohomology of moduli stacks of hyperelliptic curves. It was withdrawn because of a gap that lies in the core arguments of Theorem 1.8 and Theorem 1.13. On the other hand, Park et al. (2024b) studies projective hypersurfaces with real coefficients and presents an explicit description of the complex conjugation of singular cohomology in terms of algebraic de Rham cohomology. The authors' withdrawal note on arXiv suggests that a hidden assumption on Hodge numbers was accidentally introduced in the early set-up of section 2.2, thus making Theorem 2.3 only valid under the additional assumption. We run our evaluation using OpenAI o3 and Gemini 2.5 Pro over the two pre-prints, once for each combination.

- **Section 2.3, proof of Theorem 2.3:** They set

$$C\big(\varphi_C(U_k), \varphi_C(U_k)\big) = \int_X \varphi_C(U_k) \wedge \varphi_C(U_k)$$

  and claim it equals the nonzero quantity $U_{k,\gamma}$; however, for even $n$, $\varphi_C(U_k)$ has odd degree $n-1$, so $\varphi_C(U_k) \wedge \varphi_C(U_k) = 0$ identically, making the equality impossible and invalidating Lemma 2.5 and Theorem 2.3.

- **Example 2.4:** Using the erroneous formula without complex conjugation yields $\mu = 0$, which would force $\varphi_C([1]) = 0$, contradicting that $[1]$ represents the holomorphic 1-form. This example exposes the same flaw as in the main proof.

- **Theorem 2.3 (implicit assumption):** Formula (2.10) divides by $U_{k,\gamma}$, but the proof provides no argument that $U_{k,\gamma} \neq 0$. In fact, for odd-degree forms $U_{k,\gamma}$ vanishes, so the construction of $\tilde{f}_{ji}(x)$ is undefined.

Figure 11: o3's initial feedback on Park et al. (2024b).

- **Section 2.5:** Corollary 2.7 incorrectly describes the R-span of the algebraic representatives for the real primitive cohomology $H^{n-1}_{\mathrm{prim}}(X, \mathbb{R})$. The basis elements given, $[\omega_{j_i}]$ and $[\sqrt{-1}\,\omega_{j_i}]$, correspond under $\varphi_C$ to $\Re(\omega_k)$ and $i\,\Re(\omega_k)$. The correct basis for the real cohomology $H(X, \mathbb{R})$ should correspond to $\Re(\omega_k)$ and $\Im(\omega_k)$, where $\omega_k = \varphi_C(U_k)$. This makes the explicit description of $(\varphi_C^{-1} \circ P)\big(H^{n-1}_{\mathrm{prim}}(X, \mathbb{R})\big)$ scientifically invalid.

- **Example 2.8:** This example illustrates Corollary 2.7 for elliptic curves and thus inherits the same flaw. The stated R-span $\mathbb{R}[\ldots] + \sqrt{-1}\,\mathbb{R}[\ldots]$ describes an incorrect subspace related only to the real part of the basis forms, not the full real cohomology $H^1(X, \mathbb{R})$.

Figure 12: Gemini-2.5-Pro's initial feedback on Park et al. (2024b).

Since we have already analyzed the LLM responses for Petersen and Tommasi (2024) in Section 4, here we only describe some additional analysis on Park et al. (2024b). Our first attempt at running our pipeline introduced many OCR errors, such as missing conjugations in an integral. Both o3 and Gemini-2.5-Pro reacted to different OCR errors and identified them as mistakes (Figures 11 and 12).

After an improvement (inclusion of GPT-4.1 mentioned in Section 2), surprisingly, o3 claimed that there are no errors in Park et al. (2024b).

As to Gemini 2.5 Pro, it proposed several potential mistakes (Figure 13). First, it argued that in the statement of Theorem 2.3, "the existence of a polynomial $g_{j_i}$ such that $r_{f_{j_i}, g_{j_i}} \neq 0$" was never justified. Although this may be a fair writing feedback, this does not constitute an error. Aside from another OCR problem where the LLM saw an incorrect mathematical equation, another argument it

made was that in Example 2.8, the basis of the cohomology on the right-hand side was unconventional and $\mathbb{R} \cdot [\omega] + \mathbb{R} \cdot (i[\omega])$ was better. This was made out of context, and the LLM might have forgotten that this example was an application of Corollary 2.7.

---

- **Section 2.2, Theorem 2.3:** The theorem requires the existence of a polynomial $g_{ji}$ such that

$$r_{f_{ji}, g_{ji}} \neq 0$$

(hence $f_{ji} \notin \mathrm{Jac}(G)$). However, the paper never justifies that the chosen basis elements

$$U_k = y^i f_{ji}$$

(relative to $\mathrm{Jac}(S)$) can always be represented by polynomials $f_{ji}$ lying outside $\mathrm{Jac}(G)$. This hidden assumption is crucial for formula (2.10) to be well-defined and for the theorem to hold.

- **Section 2.2, Example 2.4:** The text claims the constant $\mu$ is proportional to

$$\left( \int_{\gamma_1} \omega \right) \left( \int_{\gamma_2} \omega \right) - \left( \int_{\gamma_2} \omega \right) \left( \int_{\gamma_1} \omega \right).$$

But by Definition 2.2 and the pairing $C(\omega, \overline{\omega})$, it should be

$$\left( \int_{\gamma_1} \omega \right) \left( \int_{\gamma_2} \overline{\omega} \right) - \left( \int_{\gamma_2} \omega \right) \left( \int_{\gamma_1} \overline{\omega} \right),$$

—the presented formula omits the complex-conjugate terms, indicating a calculation error.

- **Section 2.4:** For odd $n$, the paper defines

$$U_{k,\gamma} = PX \cdot P \cdot PX^T.$$

But since $U_{k,\gamma}$ comes from the pairing $C(\phi_C(U_k), \overline{\phi_C(U_k)})$ and $PX$ is complex, the correct expression uses the conjugate transpose:

$$U_{k,\gamma} = PX \cdot P \cdot \overline{PX^T},$$

so using $PX^T$ is mathematically incorrect.

- **Section 2.5, Example 2.8:** The basis for the real cohomology is written as

$$\mathbb{R}\big[(1 + \mu\, y^d(x))/2\big] + \sqrt{-1}\, \mathbb{R}\big[(1 + \mu\, y^d(x))/2\big],$$

which is unconventional and obscures the structure. A clearer description is

$$\mathbb{R} \cdot [\omega] + \mathbb{R} \cdot \big(i[\omega]\big),$$

explicitly showing the real subspace spanned by $[\omega]$ and its $i$-multiple.

Figure 13: Gemini-2.5-Pro's regenerated feedback on Park et al. (2024b).

## D.2 MATERIAL SCIENCE

We select Ye et al. (2025) and Altuijri et al. (2024), two recent materials-science studies. Ye et al. (2025) investigates the synthesis of covalent organic frameworks (COFs), a class of porous materials with diverse applications, using a novel superacid-catalyzed process that enables the fabrication of new types of COF architectures. Several errors were pointed out about this article, mostly due to mislabeling COFs either in the text or in the figures, but none of them impact the conclusions of the article. Altuijri et al. (2024) proposes a method to increase the hydrophobicity and electrical properties of certain types of films by incidence with low-energy argon beams. This work had a wrong caption, claiming to have used a different beam intensity on a certain panel when that panel was at a different magnification. Both these errors are easier to spot than the ones found in the mathematics articles. While some of the errors in Ye et al. (2025) require some domain knowledge in COFs, most of the other errors can be spotted by paying close attention to the text and figures.

- **Section 2.2, Model Reactions (Figures 1C–G):** Quantitative yields of individual species are derived from a single UV absorbance at 240 nm without extinction-coefficient calibration or spectral deconvolution, even though all species overlap strongly in this region. The reported 92 %, 82 %, etc. compositions are therefore not experimentally supported.

- **Section 2.2 / Figure 1H:** Assumes acetals are unaffected by $NaHCO_3$ neutralization; no control provided. If acetals hydrolyze under basic conditions, the calculated "protected" fractions are invalid, undermining the mechanistic interpretation.

- **Section 2.3:** Conclusion that barriers < 20 kcal $mol^{-1}$ imply "full reversibility at room temperature" ignores solvent and entropic contributions; a 19 kcal mol$^{-1}$ barrier can correspond to multi-hour timescales. The stated kinetic implication is not justified by the DFT data.

- **Abstract & Section 2.5:** Optimal TfOH concentration is stated as 0.006–0.016 M, yet all optimization and "standard" syntheses use 0.2 M TfOH. Internal inconsistency invalidates the universality of the claimed optimum.

- **Section 2.4 / Figure 2:** Higher PXRD peak counts are used as evidence of greater crystallinity, but raw intensity is instrument- and packing-dependent. Without normalization or an internal standard, counts cannot be compared between samples, so the claim of superior crystallinity is unsupported.

- **Section 2.7, "nine new COFs" claim:** Authors conclude AcOH "cannot" give these COFs after testing only a single AcOH condition per monomer pair. Lack of systematic screening means the impossibility claim is unsubstantiated.

- **Section 2.11, Photocatalysis:** Performance deemed "exceptional" based solely on DPBF bleaching time, with no quantum yield, turnover number, or photon-normalized rate. Comparisons to literature therefore lack a common metric, invalidating the superiority claim.

Figure 14: o3's regenerated feedback on Ye et al. (2025).

Most of the errors OpenAI's o3 pointed out on Ye et al. (2025) were on the experimental setup and the conclusions that the authors derived from them. Although the specific points raised by o3 seem to be plausible mistakes, all of them either stem from misunderstandings by o3 or from ignoring information. For example in Figure 14, the claim that the UV-derived yields in Section 2.2 (Figures 1C–G) lack support is a hallucination: the authors validate those compositions with 1H NMR spectroscopy, which produces results similar to the UV measurements, so it is incorrect to argue that the yields are unsupported. In Section 2.3, o3 contends that barriers below 20 kcal mol$^{-1}$ cannot guarantee "full reversibility" at room temperature. While entropic factors do affect reaction rates, barriers under 20 kcal mol$^{-1}$ are generally regarded as low enough for reversibility, and multi-hour timescales still allow dynamic equilibrium. This error likely stems from o3's unfamiliarity with practical conventions in kinetic chemistry. Finally, for photocatalysis performance (Section 2.11), o3 questions the "exceptional" activity because no quantum yields or turnover numbers are provided. However, o3 does not have access to the supplementary information and thus is hallucinating that no DPBF bleaching comparisons exist, Table S10 demonstrates that these COFs outperform those in the literature.

Gemini 2.5 Pro also misunderstand the authors, in it feedback regarding Section 2.7 (Figure 4C) is related to the fact that the authors claim that a certain reagent is not suitable for the production of a

- **Section 2.3 and Figure 1I (related to Section 2.2(2)):** The DFT-calculated relative energy of the acetal product P1 (+7.4 kcal/mol compared to benzaldehyde) indicates it is thermodynamically highly unfavorable, directly contradicting the experimental 81 % yield reported in Section 2.2(2), which would require a negative ΔG (approx. –0.85 kcal/mol). This discrepancy undermines the validity of the DFT calculations as an explanation for the experimentally observed acetal protection.

- **Section 2.5(1) and Figures 3A, 3C, 3E:** The text states that low-boiling point solvents (MeOH, EtOH, n-propanol) are 'unsuitable' for reactions at 120 °C due to their boiling points being below this temperature. However, Figure 3 A,C,E present PXRD data for COFs synthesized in these solvents, presumably at 120 °C. The manuscript fails to specify whether these reactions were performed in sealed vessels. If not, solvent evaporation would lead to uncontrolled and non-reproducible reaction conditions, invalidating the results obtained with these specific solvents.

- **Section 2.6:** The BET surface area for the scaled-up TFPPy-PDA-COF is reported as "1606 cm$^2$ g$^{-1}$." The correct unit is m$^2$/g. This unit error misrepresents the surface area value by a factor of 10,000 and is a fundamental error in data reporting.

- **Abstract; Introductory paragraph; Section 2.7; Figure 4C:** The paper claims "nine new imine-linked frameworks," but Figure 4C highlights ten COFs (#5, #9, #13, #14, #15, #18, #23, #24, #25, #28) as newly synthesized or inaccessible by traditional methods, creating a numerical inconsistency.

- **Unnumbered paragraph on PXRD for Figure 6H:** There is a contradiction in the indexing of PXRD peaks for TAPPy-TFPPy-COF (Figure 6H). The peaks are initially assigned to facets including (020): 'TAPPy-TFPPy-COF displayed peaks at 5.00°, 6.90°, 10.08°, 15.24°, and 23.22°...assigned to the (110), (020), (220), (330), and (001) facets...'. A subsequent concluding sentence for this group of COFs states related peaks 'were attributed to the (100), (110), (220), (330), and (001) facets...'. The differing indices for the second peak ((020) vs. (100)) for the same data create ambiguity and undermine the reliability of the structural characterization from PXRD for this COF.

Figure 15: Gemini-2.5-Pro's regenerated feedback on Ye et al. (2025).

COF and they show how it does not produce such a good results, because of this Gemini claims that the reagent can actually be used because the authors used it in a figure. Contrary to o3, it does find a real mistake by noticing that one of the characterizations of a certain COF contained the wrong unit, making the measurement 10000 times smaller.

- **Methods – SRIM simulation / Figure 2:** Authors discuss ion penetration up to 1 μm for 4 keV Ar$^+$, whereas SRIM and the plotted axis (0–1000 Å) show a stopping range below 20 nm. Conclusions about bulk (μm-scale) modification rest on an impossible depth estimate.

- **Results – EDX (Figures 3 & 4):** Claim that irradiation changed elemental composition is based on C, O, N shifts under 2 wt %, well inside normal EDX error; no statistics provided, so the asserted chemical change is unsupported.

- **Results – Contact angle / Work of adhesion (Eq. [1], Figures 7 & 8):** Reported $W_a$ values do not satisfy $W_a = \gamma_\ell(1 + \cos\theta)$; for example, $\theta = 62.1°$ for water gives 106 mJ m$^{-2}$, but the table lists 74.2 mJ m$^{-2}$. All subsequent surface-energy conclusions use these erroneous numbers.

- **Results – Surface free energy (Figure 9):** Owens–Wendt requires two liquids to solve for $\gamma_s^{\mathrm{d}}$ and $\gamma_s^{\mathrm{p}}$. With only two liquids the solution at each fluence is unique, yet the authors report independent trends for both components, indicating a mathematical inconsistency.

- **Results – Electrical conductivity (Figure 10, text):** Two mutually contradictory pristine conductivities are given ($3.8 \times 10^{-7}$ S cm$^{-1}$ vs. $1.1 \times 10^{-8}$ S cm$^{-1}$). The inconsistency invalidates the reported enhancements due to irradiation.

- **Results – Potential barrier calculation (Figure 11, Eq. [4]):** Formula $W_m = -4k_BT/m$ sets an upper magnitude of 0.104 eV at 300 K, yet values up to 1.07 eV are reported and the sign is ignored; the dielectric analysis is therefore mathematically impossible.

Figure 16: o3's regenerated feedback on Altuijri et al. (2024).

Figure 16, are feedbacks of o3 on Altuijri et al. (2024). Here, it seems like many of o3's flagged issues arise from misinterpretation rather than true errors. Its EDX objection ignores that the authors corroborate composition changes with complementary measurements and that small weight-percent shifts demand statistical treatment. Regarding contact angle, o3 uses diiodomethane data instead of water, miscomputing $W_a$. The supposed Owens–Wendt inconsistency stems from a misunderstanding that two liquids yield a unique solution per fluence, so the reported parallel trends do not imply mathematical error. Finally, the potential barrier formula comes from literature conventions, and Figure 11 presents a related quantity rather than misapplying the sign.

- **Figure 2a:** The text states SRIM simulation data is shown for depths of 0 to 1 μm, but the x-axis of Figure 2a ("Target Depth") only extends to 1000 Å (0.1 μm). This misrepresents the range of simulation data presented visually.

- **Results and Discussion – Contact Angle, Work of Adhesion, Surface Free Energy (Figures 7, 8, 9):** Table I (containing contact angle and work of adhesion data) and Table II (containing surface free energy data) are referenced but not provided. This prevents independent verification of the data plotted in Figures 7–9 and the calculations derived from them, undermining the scientific validity of these results.

- **Results and Discussion – Work of Adhesion (Figure 8):** The work of adhesion $W_a$ values reported in the text (and presumably plotted in Figure 8) are inconsistent with calculations based on contact angle data from Figure 7 and standard liquid surface tension values (e.g., water at $4 \times 10^{14}$ ions/cm$^2$: reported $W_a = 74.2 \, \mathrm{mJ/m^2}$, calculated $W_a \approx 106.4 \, \mathrm{mJ/m^2}$) . The liquid surface tension values ($\gamma_\ell$) used for calculations are not stated, and the discrepancy suggests significant errors in calculation or data, invalidating the reported $W_a$ values.

- **Results and Discussion – Figure 10 and related text:** AC conductivity $\sigma_{\mathrm{ac}}$ values at 50 Hz reported in the text for irradiated samples differ significantly (e.g. for $4 \times 10^{14}$ ions·cm$^{-2}$: text $3.8 \times 10^{-7}$ S cm$^{-1}$ vs. Fig. 10 data $\sim 3.74 \times 10^{-8}$ S cm$^{-1}$) in a summary paragraph preceding the electrical characterization. This contradiction undermines the baseline characterization of the material.

- **Results and Discussion – Electrical Properties:** The AC conductivity $\sigma_{\mathrm{ac}}$ for unirradiated (pristine) CA/PANI at 50 Hz is reported inconsistently: as $1.1 \times 10^{-8}$ S cm$^{-1}$ in the Figure 10 discussion (matching the figure) and as $3.8 \times 10^{-7}$ S cm$^{-1}$ in a preceding summary paragraph.

- **Results and Discussion – Figure 11 and Equation [4]:** The exponent $m$ for calculating $W_m$ is derived from a narrow low-frequency range in Figure 11 ($\ln \omega \approx 5.8$–$6.5$), which is not justified and may not be representative of the model used. The validity of conclusions based on $W_m$ is therefore questionable

- **Results and Discussion – Equation [4]:** The formula

$$W_m = -\frac{4k_B T}{m}$$

used for calculating the maximum potential barrier is unconventional for hopping conduction models, and its scientific basis or appropriateness for this system is not established via the provided citation [40] or derivation. Using an potentially incorrect or unsubstantiated formula invalidates the calculated $W_m$ values.

- **Results and Discussion – Calculation of $W_m$:** The text states that the estimated $W_m$ "decreased from 0.116 eV. . . to 1.03 eV." However, the authors' reported values (0.116, 0.11, 1.07, 1.03) eV show a non-monotonic behavior (slight decrease, then large increase, then slight decrease). This is a misrepresentation of their calculated results.

Figure 17: Gemini-2.5-Pro's regenerated feedback on Altuijri et al. (2024).

In Figure 17 we observe multiple errors where Gemini-2.5-Pro fails to identify visual inputs correctly; for instance, it claims that Tables I and II referenced in the discussion of Figures 7–9 are missing, even though those tables are present, indicating it has failed to identify the context. Additionally, in Figure 10, it misreads a log-scaled plot as linear and thereby hallucinates values.

- **Section 3.1.1 / Table 1:** Temperature "predictions" obtained from $T = \frac{PM}{\rho R}$ are tautological because $\rho = \frac{PM}{RT}$ by definition; using measured $P$, $M$, and $\rho$ always reproduces the measured $T$. This provides no evidence against the greenhouse effect yet is used as the cornerstone of the paper's conclusions.

- **Section 3.1.2 (Equations 3–4 and Figure 1):** The forcing equation is applied with $\log_{10}$ instead of the natural log, reducing $\Delta F$ by $\approx 2.3\times$ and yielding a spurious warming of $0.64$–$0.81\,^\circ\mathrm{C}$ for $CO_2$ doubling. Correct use of $\ln$ gives $1.5$–$1.9\,^\circ\mathrm{C}$, invalidating the claim that future warming is negligible.

- **Section 3.1.2 (Climate-sensitivity discussion):** Derived ECS $\approx 0.5\,^\circ\mathrm{C}$ ignores water-vapour, lapse-rate, and cloud feedbacks without justification; hence the stated upper bound on warming lacks the physical components known to amplify $CO_2$ forcing.

- **Section 3.1.4 (Residence time of $CO_2$):** Paper conflates $3$–$5$ yr molecular turnover time with the multi-century adjustment time of a concentration perturbation, leading to the unsupported claim that anthropogenic $CO_2$ 'quickly disappears' and is climatically irrelevant.

- **Section 3.1.4 (Anthropogenic share argument):** Using the fact that anthropogenic emissions are $\sim 4\%$ of gross annual flux to assert they cannot raise atmospheric $CO_2$ ignores that natural sources and sinks are nearly balanced; failure to perform a mass-balance analysis renders the conclusion scientifically invalid.

- **Section 3.1 (General claim of "no greenhouse effect"):** Assertion that adiabatic compression or thermal conductivity alone explains the 33 K surface–space temperature difference disregards radiative-transfer energy balance and provides no quantitative mechanism, contradicting well-established physics without offering an alternative that reproduces observed fluxes.

Figure 18: o3's regenerated feedback on Simpson (2024).

### D.3 Environmental Science

We selected Simpson (2024) in environmental science, the paper conductsa comprehensive evaluation of human contributions to atmospheric $CO_2$. In a data-driven analysis, Simpson (2024) argues that human-induced $CO_2$ emissions are negligible compared to natural sources, thereby questioning the validity of the Greenhouse Gas Hypothesis. However, the original version of the paper contains a miscalculation in Section 3.1.2, "Measurement of Infrared Absorption of the Earth's Atmosphere." Specifically, the author incorrectly applies Equation (3) from the IPCC: $F = 5.35 \ln\left(\frac{C_t}{C_0}\right)$ by using the base-10 logarithm ($\log_{10}$) instead of the natural logarithm ($\ln$), leading to erroneous numerical values. Regarding the miscalculation, in Figure 18, o3 correctly locates the target error: "$\log_{10}$ instead of the natural log, reducing $\Delta F$ by $\approx 2.3\times$ and yielding a spurious warming of $0.64$–$0.81\,^\circ\mathrm{C}$ for $CO_2$ doubling. Correct use of $\ln$ gives $1.5$–$1.9\,^\circ\mathrm{C}$," highlighting a mismatch between the scale of results and the correct calculation basis. However, its subsequent claim that this error "invalidates the assertion of negligible future warming" seems overstated. As the authors acknowledge, projected temperature increases remain modest. In short, the magnitude of these miscalculations is insufficient to overturn the paper's broader argument about limited warming.

On the other hand, Gemini2.5 fails to point out the specific error.

## E Additional Details on Spot

**License & Copyright** SPOT comprises 83 manuscripts published across 28 venues (including arXiv). Of these, 62 (74.7 %) are openly accessible under a CC license; we publicly share our fully preprocessed versions via the Hugging Face Hub. The remaining 21 (25.3 %) are paywalled, so we do not redistribute them directly. Organizations with institutional access to Springer Nature or Elsevier can apply our preprocessing pipeline to generate their own versions.

**Date** To minimize contamination against parametric knowledge (Bejan et al., 2023), we aim to include only papers published from 2024 onward. As Figure 19 shows, the bulk of our corpus dates

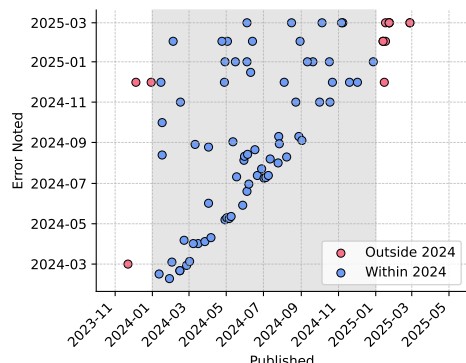

Figure 19: **Publication dates against first error-notice dates for the 83 manuscripts.** Each point denotes one paper; blue markers note papers published in 2024, while red markers are those otherwise.

to 2024, with ten papers from 2025 and three that originally appeared before 2023. Those three early manuscripts passed our automated filters because revisions were submitted after 2024; we retained them since their first error notices appeared in March 2024, minimizing any chance that models were exposed to the original withdrawal details during training.

**Annotation** We use human annotators in Section 2 during the benchmark creation process. Details on the annotator guideline are available in Figure 20, a sample image of the platform in Figure 21.

## F ADDITIONAL DETAILS ON EVALUATION

Evaluation consists of two phases. In the first phase, the target LLM is prompted to identify potential errors in each paper using our "Generation Prompt." In the second phase, we employ GPT-4.1 to align and compare the model's candidates against the ground-truth annotations with our "Evaluation Prompt." In the remainder of this section, we specify details on generation configurations and present the full text of both prompts.

### F.1 GENERATION CONFIGURATIONS

For each model, we adopt the provider's recommended parameters when available; otherwise, we use a sampling temperature of 0.6, top-p of 0.95, a repetition penalty of 1.0, and enforce a minimum of 8 and a maximum of 8192 tokens.

A lightweight Streamlit app for labeling errors discussed on **PubPeer** or papers **withdrawn from arXiv**. Contributors review randomly selected papers, answer guided questions, and append their work to `annotations.csv`.

GETTING STARTED

PREREQUISITES

- Python $\geq 3.8$
- Streamlit
- pandas

INSTALLATION

```
# 1 Clone the repo
git clone https://github.com/guijinSON/ai4s_r2.git
cd ai4s_r2

# 2 Install dependencies
pip install streamlit pandas
```

DATASET

`retracted_machine_filtered_final.csv` ships with the repository—no additional download required.

USAGE

```
streamlit run streamlit_sample.py
```

1. **Shuffle Sample** loads a random paper.
2. Complete the six annotation questions in the right panel.
3. Click **Save Annotation** to append to `annotations.csv`.
4. Repeat until 3–5 rows are completed.
5. Click **Submit** to send `annotations.csv` to the maintainer.

Figure 20: Guideline provided to annotators.

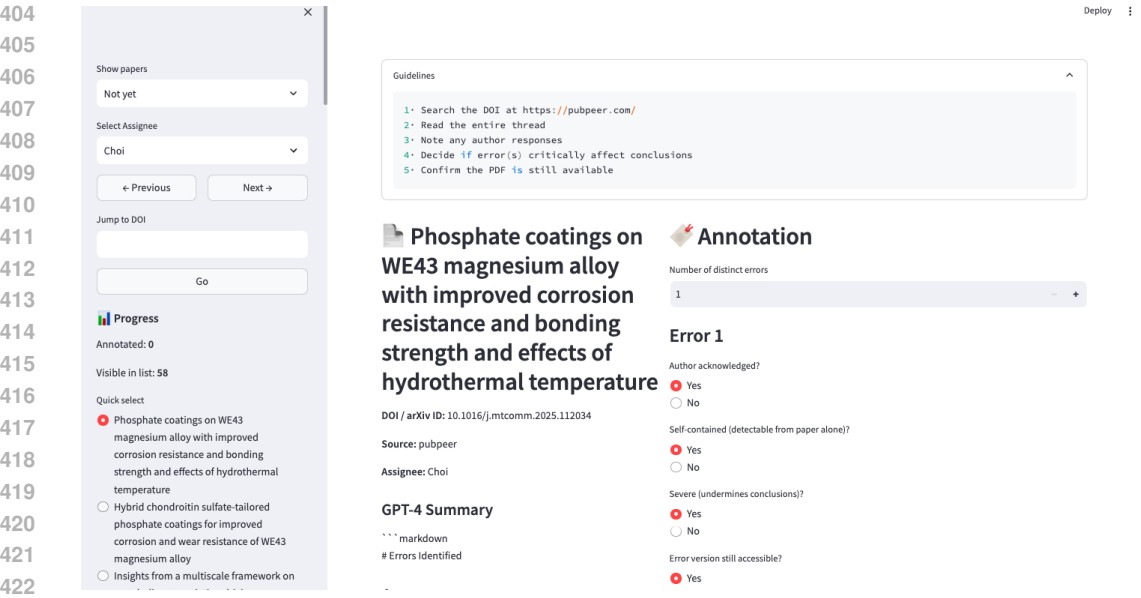

Figure 21: Example image of annotation platform.

## F.2 PROMPTS

**Generation Prompt**

You are a **scientific-rigor auditor**. You will receive the parsed contents of a research paper. Your job is to identify only those errors or flaws that directly undermine the scientific validity of the paper's methods, analyses, or conclusions. Your sole focus is identifying flaws, such as errors in experimental design, data integrity, calculations, statistical inference, or reproducibility, that directly call into question the validity of a specific claim, paragraph, or the paper. Do not report issues purely presentational, rhetorical, stylistic, or related to citation practices.

—

After you've done a **detailed walkthrough** of the paper, output exactly in this format—no extra keys or commentary:

```
‘‘‘
<analysis>
{detailed walk-through of how you checked each section/figure and why you flagged (or did not flag) any flaw}
</analysis>

<response>

{
    "has_error": <true | false>,
    "errors": [
        {
            "location": "Section 2.1",
            "description": "Claim that 'all X are Y' is ..."
        },
        {
            "location": "Figure 3",
            "description": "X-Axis labeled 'Time (s)' but units ..."
        }
        // ...more entries...
    ]
}

</response>
```

- Do not include other keys or prose outside these two tagged blocks.
- Do not report stylistic or citation issues.

**Evaluation Prompt**

You are an expert LLM-as-a-Judge. You will receive a JSON object with two arrays:

1. "annotations": the ground-truth errors (each has "location" and "description").
2. "predictions": the model's reported errors (same format).

**Task**
1. Compare each prediction against each annotation.
2. A match occurs only when both "location" and "description" are identical.
3. Your output should be generated in the following format:

<analysis>
Analysis and comparison of each prediction and annotation.
</analysis>
<response>

```
{
   "matches": [
        {
        "location": the location of the matched object, which should be
        based on the annotated location,
        "description": your explanation on why you think it is a match.
        },
          {
        "location": ... ,
        "description": ...
        },
   ]
}
```

</response>

Be rigorous in considering matches; the location may be slightly differently named, but the description must match overall.

# G   DETAILED RESULTS

1. In Table 4 to 13 we present detailed results of each model from Table 2.

2. In Table 14 to 26 we present detailed results of the text-only evaluation from Table 3.

Table 4: Mean and standard deviation of pass@$K$ for o3 ($K \in \{1, 2, 4\}$) by error category (left) and paper category (right). Detailed evaluations results for Table 2.

| Error Category | | | | Paper Category | | | |
| --- | --- | --- | --- | --- | --- | --- | --- |
| Category | pass@1 | pass@2 | pass@4 | Category | pass@1 | pass@2 | pass@4 |
| Data Inconsistency | $13.1_{8.3}$ | $19.4_{7.0}$ | $25.7_{4.3}$ | Biology | $5.1_{5.8}$ | $9.4_{5.9}$ | $14.5_{2.5}$ |
| Equation / proof | $33.6_{3.8}$ | $51.6_{6.1}$ | $67.5_{3.7}$ | Chemistry | $0.0_{0.0}$ | $0.0_{0.0}$ | $0.0_{0.0}$ |
| Experiment setup | $0.0_{0.0}$ | $0.0_{0.0}$ | $0.0_{0.0}$ | Computer Science | $21.0_{11.5}$ | $36.1_{12.1}$ | $55.9_{9.5}$ |
| Figure duplication | $0.0_{0.0}$ | $0.0_{0.0}$ | $0.0_{0.0}$ | Engineering | $0.0_{0.0}$ | $0.0_{0.0}$ | $0.0_{0.0}$ |
| Reagent identity | $22.0_{25.1}$ | $40.7_{25.5}$ | $62.7_{10.8}$ | Environmental Science | $5.1_{12.0}$ | $10.7_{15.6}$ | $22.1_{15.8}$ |
| Statistical reporting | $45.7_{17.2}$ | $62.8_{17.9}$ | $88.4_{12.5}$ | Materials Science | $14.2_{6.0}$ | $16.7_{0.0}$ | $16.7_{0.0}$ |
| | | | | Mathematics | $34.3_{5.3}$ | $53.3_{6.0}$ | $67.6_{2.5}$ |
| | | | | Medicine | $0.0_{0.0}$ | $0.0_{0.0}$ | $0.0_{0.0}$ |
| | | | | Multidisciplinary | $20.0_{0.0}$ | $20.0_{0.0}$ | $20.0_{0.0}$ |
| | | | | Physics | $33.7_{20.7}$ | $56.3_{17.6}$ | $79.0_{7.1}$ |

Table 5: Mean and standard deviation of pass@$K$ for GPT-4.1 ($K \in \{1, 2, 4\}$) by error category (left) and paper category (right). Detailed evaluations results for Table 2.

| Error Category | | | | Paper Category | | | |
| --- | --- | --- | --- | --- | --- | --- | --- |
| Category | pass@1 | pass@2 | pass@4 | Category | pass@1 | pass@2 | pass@4 |
| Data Inconsistency | $6.4_{5.6}$ | $11.4_{6.6}$ | $19.2_{6.1}$ | Biology | $21.4_{6.5}$ | $34.8_{7.5}$ | $48.9_{7.3}$ |
| Equation / proof | $0.4_{1.0}$ | $0.8_{1.3}$ | $1.5_{1.5}$ | Chemistry | $2.1_{5.5}$ | $4.0_{7.1}$ | $8.2_{8.3}$ |
| Experiment setup | $0.0_{0.0}$ | $0.0_{0.0}$ | $0.0_{0.0}$ | Computer Science | $0.0_{0.0}$ | $0.0_{0.0}$ | $0.0_{0.0}$ |
| Figure duplication | $16.3_{4.9}$ | $27.6_{5.7}$ | $41.1_{5.6}$ | Engineering | $12.9_{21.9}$ | $23.1_{24.9}$ | $39.3_{20.5}$ |
| Reagent identity | $4.5_{11.4}$ | $8.6_{14.6}$ | $16.5_{16.7}$ | Environmental Science | $0.0_{0.0}$ | $0.0_{0.0}$ | $0.0_{0.0}$ |
| Statistical reporting | $0.0_{0.0}$ | $0.0_{0.0}$ | $0.0_{0.0}$ | Materials Science | $14.6_{9.8}$ | $26.8_{11.3}$ | $41.5_{9.4}$ |
| | | | | Mathematics | $0.0_{0.0}$ | $0.0_{0.0}$ | $0.0_{0.0}$ |
| | | | | Medicine | $5.9_{16.1}$ | $11.3_{20.9}$ | $25.2_{25.0}$ |
| | | | | Multidisciplinary | $12.7_{8.4}$ | $22.9_{8.7}$ | $36.4_{7.4}$ |
| | | | | Physics | $0.0_{0.0}$ | $0.0_{0.0}$ | $0.0_{0.0}$ |

Table 6: Mean and standard deviation of pass@$K$ for Gemini-2.5-Pro ($K \in \{1, 2, 4\}$) by error category (left) and paper category (right). Detailed evaluations results for Table 2.

| Error Category | | | | Paper Category | | | |
| --- | --- | --- | --- | --- | --- | --- | --- |
| Category | pass@1 | pass@2 | pass@4 | Category | pass@1 | pass@2 | pass@4 |
| Data Inconsistency | $12.6_{7.0}$ | $22.3_{7.6}$ | $36.5_{6.9}$ | Biology | $2.0_{3.4}$ | $3.9_{4.4}$ | $7.6_{5.1}$ |
| Equation / proof | $11.8_{7.0}$ | $21.9_{7.9}$ | $38.4_{7.3}$ | Chemistry | $8.6_{8.3}$ | $15.5_{9.5}$ | $25.9_{8.7}$ |
| Experiment setup | $0.0_{0.0}$ | $0.0_{0.0}$ | $0.0_{0.0}$ | Computer Science | $8.2_{7.1}$ | $16.7_{9.7}$ | $31.7_{11.5}$ |
| Figure duplication | $1.5_{2.6}$ | $2.8_{3.4}$ | $5.5_{3.9}$ | Engineering | $0.0_{0.0}$ | $0.0_{0.0}$ | $0.0_{0.0}$ |
| Reagent identity | $0.0_{0.0}$ | $0.0_{0.0}$ | $0.0_{0.0}$ | Environmental Science | $8.3_{14.4}$ | $15.6_{16.6}$ | $26.4_{13.6}$ |
| Statistical reporting | $15.2_{24.3}$ | $31.6_{31.5}$ | $57.7_{32.0}$ | Materials Science | $3.9_{7.1}$ | $7.5_{8.3}$ | $13.0_{6.9}$ |
| | | | | Mathematics | $11.3_{8.5}$ | $21.1_{9.7}$ | $38.0_{8.6}$ |
| | | | | Medicine | $0.0_{0.0}$ | $0.0_{0.0}$ | $0.0_{0.0}$ |
| | | | | Multidisciplinary | $11.2_{5.9}$ | $20.3_{7.7}$ | $32.8_{8.8}$ |
| | | | | Physics | $14.6_{14.5}$ | $25.7_{14.9}$ | $39.7_{12.8}$ |

Table 7: Mean and standard deviation of $\text{pass}@K$ for Gemini-2.0-Flash-Lite-001 ($K \in \{1, 2, 4\}$) by error category (left) and paper category (right). Detailed evaluations results for Table 2.

| Error Category | | | | Paper Category | | | |
|---|---|---|---|---|---|---|---|
| Category | pass@1 | pass@2 | pass@4 | Category | pass@1 | pass@2 | pass@4 |
| Data Inconsistency | $1.8_{3.1}$ | $3.5_{4.0}$ | $7.0_{4.7}$ | Biology | $3.7_{3.8}$ | $7.6_{4.9}$ | $15.4_{5.7}$ |
| Equation / proof | $0.0_{0.0}$ | $0.0_{0.0}$ | $0.0_{0.0}$ | Chemistry | $2.1_{5.5}$ | $4.2_{7.2}$ | $8.1_{8.3}$ |
| Experiment setup | $0.0_{0.0}$ | $0.0_{0.0}$ | $0.0_{0.0}$ | Computer Science | $0.0_{0.0}$ | $0.0_{0.0}$ | $0.0_{0.0}$ |
| Figure duplication | $3.2_{2.9}$ | $6.3_{3.7}$ | $12.9_{4.2}$ | Engineering | $0.0_{0.0}$ | $0.0_{0.0}$ | $0.0_{0.0}$ |
| Reagent identity | $3.9_{10.7}$ | $8.7_{14.7}$ | $16.9_{16.7}$ | Environmental Science | $0.0_{0.0}$ | $0.0_{0.0}$ | $0.0_{0.0}$ |
| Statistical reporting | $0.0_{0.0}$ | $0.0_{0.0}$ | $0.0_{0.0}$ | Materials Science | $2.1_{5.5}$ | $4.1_{7.2}$ | $8.3_{8.3}$ |
| | | | | Mathematics | $0.0_{0.0}$ | $0.0_{0.0}$ | $0.0_{0.0}$ |
| | | | | Medicine | $3.1_{8.2}$ | $6.6_{11.0}$ | $12.6_{12.5}$ |
| | | | | Multidisciplinary | $3.7_{4.8}$ | $7.2_{6.3}$ | $14.9_{7.4}$ |
| | | | | Physics | $0.0_{0.0}$ | $0.0_{0.0}$ | $0.0_{0.0}$ |

Table 8: Mean and standard deviation of $\text{pass}@K$ for Claude-3.7-Sonnet:Thinking ($K \in \{1, 2, 4\}$) by error category (left) and paper category (right). Detailed evaluations results for Table 2.

| Error Category | | | | Paper Category | | | |
|---|---|---|---|---|---|---|---|
| Category | pass@1 | pass@2 | pass@4 | Category | pass@1 | pass@2 | pass@4 |
| Data Inconsistency | $8.0_{4.2}$ | $15.7_{5.8}$ | $29.3_{6.9}$ | Biology | $13.4_{7.3}$ | $23.9_{9.9}$ | $41.2_{10.6}$ |
| Equation / proof | $0.8_{1.3}$ | $1.6_{1.8}$ | $3.0_{2.0}$ | Chemistry | $2.1_{5.5}$ | $4.1_{7.2}$ | $8.3_{8.3}$ |
| Experiment setup | $0.0_{0.0}$ | $0.0_{0.0}$ | $0.0_{0.0}$ | Computer Science | $0.0_{0.0}$ | $0.0_{0.0}$ | $0.0_{0.0}$ |
| Figure duplication | $10.6_{3.5}$ | $19.7_{4.9}$ | $34.9_{6.0}$ | Engineering | $6.2_{16.5}$ | $12.0_{21.4}$ | $24.8_{25.0}$ |
| Reagent identity | $3.9_{10.8}$ | $7.5_{13.9}$ | $16.8_{16.7}$ | Environmental Science | $16.4_{23.2}$ | $33.1_{29.8}$ | $59.8_{29.3}$ |
| Statistical reporting | $3.1_{8.3}$ | $6.1_{10.7}$ | $12.5_{12.5}$ | Materials Science | $10.4_{11.4}$ | $20.8_{14.3}$ | $38.4_{14.5}$ |
| | | | | Mathematics | $0.6_{1.6}$ | $1.3_{2.2}$ | $2.5_{2.5}$ |
| | | | | Medicine | $6.2_{10.8}$ | $12.1_{14.2}$ | $24.9_{16.4}$ |
| | | | | Multidisciplinary | $10.0_{10.0}$ | $19.2_{13.2}$ | $35.8_{14.9}$ |
| | | | | Physics | $0.0_{0.0}$ | $0.0_{0.0}$ | $0.0_{0.0}$ |

Table 9: Mean and standard deviation of $\text{pass}@K$ for Claude-3.7-Sonnet ($K \in \{1, 2, 4\}$) by error category (left) and paper category (right). Detailed evaluations results for Table 2.

| Error Category | | | | Paper Category | | | |
|---|---|---|---|---|---|---|---|
| Category | pass@1 | pass@2 | pass@4 | Category | pass@1 | pass@2 | pass@4 |
| Data Inconsistency | $7.0_{6.2}$ | $13.6_{7.7}$ | $25.4_{8.1}$ | Biology | $10.6_{5.3}$ | $17.8_{5.5}$ | $27.3_{5.0}$ |
| Equation / proof | $0.4_{1.0}$ | $0.7_{1.3}$ | $1.5_{1.5}$ | Chemistry | $4.2_{7.2}$ | $8.2_{9.3}$ | $16.4_{10.9}$ |
| Experiment setup | $0.0_{0.0}$ | $0.0_{0.0}$ | $0.0_{0.0}$ | Computer Science | $0.0_{0.0}$ | $0.0_{0.0}$ | $0.0_{0.0}$ |
| Figure duplication | $8.8_{4.8}$ | $15.5_{4.8}$ | $25.1_{4.0}$ | Engineering | $0.0_{0.0}$ | $0.0_{0.0}$ | $0.0_{0.0}$ |
| Reagent identity | $8.6_{14.6}$ | $15.4_{16.6}$ | $26.1_{13.8}$ | Environmental Science | $3.9_{10.8}$ | $7.5_{13.9}$ | $16.8_{16.7}$ |
| Statistical reporting | $0.0_{0.0}$ | $0.0_{0.0}$ | $0.0_{0.0}$ | Materials Science | $8.2_{8.3}$ | $15.9_{10.2}$ | $30.1_{11.2}$ |
| | | | | Mathematics | $0.0_{0.0}$ | $0.0_{0.0}$ | $0.0_{0.0}$ |
| | | | | Medicine | $3.4_{8.5}$ | $6.4_{10.9}$ | $12.8_{12.5}$ |
| | | | | Multidisciplinary | $13.8_{11.2}$ | $25.4_{12.7}$ | $42.6_{10.9}$ |
| | | | | Physics | $0.0_{0.0}$ | $0.0_{0.0}$ | $0.0_{0.0}$ |

Table 10: Mean and standard deviation of $\text{pass}@K$ for Qwen2.5-VL-72B-instruct ($K \in \{1, 2, 4\}$) by error category (left) and paper category (right). Detailed evaluations results for Table 2.

| Error Category | | | | Paper Category | | | |
|---|---|---|---|---|---|---|---|
| Category | pass@1 | pass@2 | pass@4 | Category | pass@1 | pass@2 | pass@4 |
| Data Inconsistency | $0.4_{1.7}$ | $0.9_{2.4}$ | $1.8_{3.1}$ | Biology | $1.5_{3.1}$ | $3.0_{4.1}$ | $6.2_{5.7}$ |
| Equation / proof | $0.0_{0.0}$ | $0.0_{0.0}$ | $0.0_{0.0}$ | Chemistry | $0.9_{3.9}$ | $2.1_{5.5}$ | $4.3_{7.3}$ |
| Experiment setup | $0.0_{0.0}$ | $0.0_{0.0}$ | $0.0_{0.0}$ | Computer Science | $0.0_{0.0}$ | $0.0_{0.0}$ | $0.0_{0.0}$ |
| Figure duplication | $1.0_{1.6}$ | $1.9_{2.2}$ | $3.9_{3.0}$ | Engineering | $3.1_{12.1}$ | $6.6_{16.9}$ | $12.8_{21.8}$ |
| Reagent identity | $0.0_{0.0}$ | $0.0_{0.0}$ | $0.0_{0.0}$ | Environmental Science | $0.0_{0.0}$ | $0.0_{0.0}$ | $0.0_{0.0}$ |
| Statistical reporting | $0.0_{0.0}$ | $0.0_{0.0}$ | $0.0_{0.0}$ | Materials Science | $0.0_{0.0}$ | $0.0_{0.0}$ | $0.0_{0.0}$ |
| | | | | Mathematics | $0.0_{0.0}$ | $0.0_{0.0}$ | $0.0_{0.0}$ |
| | | | | Medicine | $0.0_{0.0}$ | $0.0_{0.0}$ | $0.0_{0.0}$ |
| | | | | Multidisciplinary | $0.0_{0.0}$ | $0.0_{0.0}$ | $0.0_{0.0}$ |
| | | | | Physics | $0.0_{0.0}$ | $0.0_{0.0}$ | $0.0_{0.0}$ |

Table 11: Mean and standard deviation of $\text{pass}@K$ for Qwen2.5-VL-32B-instruct ($K \in \{1, 2, 4\}$) by error category (left) and paper category (right). Detailed evaluations results for Table 2.

| Error Category | | | | Paper Category | | | |
|---|---|---|---|---|---|---|---|
| Category | pass@1 | pass@2 | pass@4 | Category | pass@1 | pass@2 | pass@4 |
| Data Inconsistency | $0.9_{2.4}$ | $1.8_{3.1}$ | $3.5_{3.6}$ | Biology | $9.7_{8.5}$ | $16.4_{8.3}$ | $25.5_{8.1}$ |
| Equation / proof | $0.0_{0.0}$ | $0.0_{0.0}$ | $0.0_{0.0}$ | Chemistry | $6.4_{11.6}$ | $12.0_{14.0}$ | $21.1_{13.3}$ |
| Experiment setup | $0.0_{0.0}$ | $0.0_{0.0}$ | $0.0_{0.0}$ | Computer Science | $0.0_{0.0}$ | $0.0_{0.0}$ | $0.0_{0.0}$ |
| Figure duplication | $4.7_{4.5}$ | $7.9_{4.9}$ | $12.2_{4.8}$ | Engineering | $0.0_{0.0}$ | $0.0_{0.0}$ | $0.0_{0.0}$ |
| Reagent identity | $8.0_{14.3}$ | $16.1_{16.7}$ | $26.6_{13.4}$ | Environmental Science | $0.0_{0.0}$ | $0.0_{0.0}$ | $0.0_{0.0}$ |
| Statistical reporting | $0.0_{0.0}$ | $0.0_{0.0}$ | $0.0_{0.0}$ | Materials Science | $0.0_{0.0}$ | $0.0_{0.0}$ | $0.0_{0.0}$ |
| | | | | Mathematics | $0.0_{0.0}$ | $0.0_{0.0}$ | $0.0_{0.0}$ |
| | | | | Medicine | $0.0_{0.0}$ | $0.0_{0.0}$ | $0.0_{0.0}$ |
| | | | | Multidisciplinary | $0.0_{0.0}$ | $0.0_{0.0}$ | $0.0_{0.0}$ |
| | | | | Physics | $0.0_{0.0}$ | $0.0_{0.0}$ | $0.0_{0.0}$ |

Table 12: Mean and standard deviation of $\text{pass}@K$ for Llama-4-Maverick ($K \in \{1, 2, 4\}$) by error category (left) and paper category (right). Detailed evaluations results for Table 2.

| Error Category | | | | Paper Category | | | |
|---|---|---|---|---|---|---|---|
| Category | pass@1 | pass@2 | pass@4 | Category | pass@1 | pass@2 | pass@4 |
| Data Inconsistency | $0.8_{2.3}$ | $1.9_{3.1}$ | $3.6_{3.6}$ | Biology | $2.9_{5.4}$ | $5.4_{6.4}$ | $9.9_{6.1}$ |
| Equation / proof | $0.0_{0.0}$ | $0.0_{0.0}$ | $0.0_{0.0}$ | Chemistry | $2.2_{5.7}$ | $4.3_{7.3}$ | $8.5_{8.3}$ |
| Experiment setup | $0.0_{0.0}$ | $0.0_{0.0}$ | $0.0_{0.0}$ | Computer Science | $0.0_{0.0}$ | $0.0_{0.0}$ | $0.0_{0.0}$ |
| Figure duplication | $1.9_{2.7}$ | $3.6_{3.2}$ | $6.6_{3.1}$ | Engineering | $0.0_{0.0}$ | $0.0_{0.0}$ | $0.0_{0.0}$ |
| Reagent identity | $4.1_{11.0}$ | $8.0_{14.2}$ | $16.5_{16.7}$ | Environmental Science | $0.0_{0.0}$ | $0.0_{0.0}$ | $0.0_{0.0}$ |
| Statistical reporting | $0.0_{0.0}$ | $0.0_{0.0}$ | $0.0_{0.0}$ | Materials Science | $4.2_{7.2}$ | $8.7_{9.7}$ | $16.7_{11.1}$ |
| | | | | Mathematics | $0.0_{0.0}$ | $0.0_{0.0}$ | $0.0_{0.0}$ |
| | | | | Medicine | $0.0_{0.0}$ | $0.0_{0.0}$ | $0.0_{0.0}$ |
| | | | | Multidisciplinary | $0.0_{0.0}$ | $0.0_{0.0}$ | $0.0_{0.0}$ |
| | | | | Physics | $0.0_{0.0}$ | $0.0_{0.0}$ | $0.0_{0.0}$ |

Table 13: Mean and standard deviation of pass@$K$ for Llama-4-Scout ($K \in \{1, 2, 4\}$) by error category (left) and paper category (right). Detailed evaluations results for Table 2.

| Error Category | | | | Paper Category | | | |
|---|---|---|---|---|---|---|---|
| Category | pass@1 | pass@2 | pass@4 | Category | pass@1 | pass@2 | pass@4 |
| Data Inconsistency | $2.6_{3.4}$ | $4.9_{4.1}$ | $9.2_{4.2}$ | Biology | $6.5_{6.0}$ | $12.8_{7.9}$ | $25.3_{9.6}$ |
| Equation / proof | $0.0_{0.0}$ | $0.0_{0.0}$ | $0.0_{0.0}$ | Chemistry | $6.1_{8.0}$ | $11.5_{9.5}$ | $21.4_{9.7}$ |
| Experiment setup | $0.0_{0.0}$ | $0.0_{0.0}$ | $0.0_{0.0}$ | Computer Science | $0.0_{0.0}$ | $0.0_{0.0}$ | $0.0_{0.0}$ |
| Figure duplication | $3.6_{4.9}$ | $7.0_{6.4}$ | $14.0_{7.6}$ | Engineering | $6.2_{16.5}$ | $12.2_{21.5}$ | $24.9_{25.0}$ |
| Reagent identity | $3.9_{10.7}$ | $8.7_{14.7}$ | $16.9_{16.7}$ | Environmental Science | $0.0_{0.0}$ | $0.0_{0.0}$ | $0.0_{0.0}$ |
| Statistical reporting | $3.1_{8.3}$ | $6.1_{10.7}$ | $12.5_{12.5}$ | Materials Science | $0.0_{0.0}$ | $0.0_{0.0}$ | $0.0_{0.0}$ |
| | | | | Mathematics | $0.0_{0.0}$ | $0.0_{0.0}$ | $0.0_{0.0}$ |
| | | | | Medicine | $3.1_{8.3}$ | $6.1_{10.7}$ | $12.5_{12.5}$ |
| | | | | Multidisciplinary | $1.2_{3.3}$ | $2.4_{4.3}$ | $5.0_{5.0}$ |
| | | | | Physics | $0.0_{0.0}$ | $0.0_{0.0}$ | $0.0_{0.0}$ |

Table 14: Mean and standard deviation of pass@$K$ for o3 ($K \in \{1, 2, 4\}$) by error category (left) and paper category (right). Detailed evaluation results for text-only evaluation of Table 3.

| Error Category | | | | Paper Category | | | |
|---|---|---|---|---|---|---|---|
| Category | pass@1 | pass@2 | pass@4 | Category | pass@1 | pass@2 | pass@4 |
| Data Inconsistency | $14.6_{5.5}$ | $25.2_{8.3}$ | $38.0_{9.7}$ | Biology | $6.2_{16.5}$ | $13.1_{22.0}$ | $25.1_{25.0}$ |
| Equation / proof | $24.6_{4.9}$ | $41.3_{5.5}$ | $62.8_{5.7}$ | Computer Science | $24.0_{7.8}$ | $42.8_{8.9}$ | $66.8_{7.5}$ |
| Experiment setup | $6.7_{17.0}$ | $12.9_{21.9}$ | $24.8_{25.0}$ | Environmental Science | $12.0_{21.4}$ | $24.1_{25.0}$ | $40.0_{20.0}$ |
| Reagent identity | $8.3_{14.4}$ | $17.1_{19.0}$ | $33.0_{21.7}$ | Materials Science | $0.0_{0.0}$ | $0.0_{0.0}$ | $0.0_{0.0}$ |
| Statistical reporting | $24.8_{17.8}$ | $44.1_{18.8}$ | $66.1_{12.4}$ | Mathematics | $21.8_{5.5}$ | $37.7_{7.9}$ | $58.9_{8.4}$ |
| | | | | Medicine | $12.4_{33.0}$ | $25.1_{43.4}$ | $48.7_{50.0}$ |
| | | | | Multidisciplinary | $33.3_{0.0}$ | $41.7_{14.5}$ | $49.6_{16.7}$ |
| | | | | Physics | $27.4_{14.4}$ | $46.0_{14.5}$ | $67.6_{10.5}$ |

Table 15: Mean and standard deviation of pass@$K$ for GPT-4.1 ($K \in \{1, 2, 4\}$) by error category (left) and paper category (right). Detailed evaluation results for text-only evaluation of Table 3.

| Error Category | | | | Paper Category | | | |
|---|---|---|---|---|---|---|---|
| Category | pass@1 | pass@2 | pass@4 | Category | pass@1 | pass@2 | pass@4 |
| Data Inconsistency | $8.6_{14.6}$ | $15.7_{16.6}$ | $26.1_{13.8}$ | Biology | $6.2_{16.5}$ | $13.1_{22.0}$ | $25.1_{25.0}$ |
| Equation / proof | $6.0_{3.4}$ | $10.8_{3.6}$ | $18.1_{3.1}$ | Computer Science | $4.1_{4.2}$ | $7.9_{5.2}$ | $14.8_{5.4}$ |
| Experiment setup | $0.0_{0.0}$ | $0.0_{0.0}$ | $0.0_{0.0}$ | Environmental Science | $19.6_{35.6}$ | $36.4_{42.2}$ | $64.6_{40.1}$ |
| Reagent identity | $8.3_{14.4}$ | $16.7_{19.1}$ | $33.2_{21.9}$ | Materials Science | $0.0_{0.0}$ | $0.0_{0.0}$ | $0.0_{0.0}$ |
| Statistical reporting | $15.6_{12.1}$ | $22.3_{7.8}$ | $25.0_{0.0}$ | Mathematics | $4.3_{3.9}$ | $8.0_{4.3}$ | $13.5_{3.8}$ |
| | | | | Medicine | $12.4_{33.0}$ | $24.0_{42.7}$ | $49.5_{50.0}$ |
| | | | | Multidisciplinary | $45.9_{16.2}$ | $71.3_{17.4}$ | $92.2_{14.1}$ |
| | | | | Physics | $0.0_{0.0}$ | $0.0_{0.0}$ | $0.0_{0.0}$ |

Table 16: Mean and standard deviation of pass@$K$ for Gemini-2.5-Pro ($K \in \{1, 2, 4\}$) by error category (left) and paper category (right). Detailed evaluation results for text-only evaluation of Table 3.

| Error Category | | | | Paper Category | | | |
|---|---|---|---|---|---|---|---|
| Category | pass@1 | pass@2 | pass@4 | Category | pass@1 | pass@2 | pass@4 |
| Data Inconsistency | $4.1_{7.2}$ | $7.7_{8.3}$ | $13.0_{6.9}$ | Biology | $0.0_{0.0}$ | $0.0_{0.0}$ | $0.0_{0.0}$ |
| Equation / proof | $7.6_{4.0}$ | $11.6_{3.9}$ | $16.2_{3.8}$ | Computer Science | $4.2_{5.9}$ | $8.2_{7.5}$ | $14.8_{8.5}$ |
| Experiment setup | $0.0_{0.0}$ | $0.0_{0.0}$ | $0.0_{0.0}$ | Environmental Science | $12.9_{21.9}$ | $23.5_{25.0}$ | $38.6_{21.0}$ |
| Reagent identity | $8.3_{14.4}$ | $15.4_{16.6}$ | $26.1_{13.8}$ | Materials Science | $0.0_{0.0}$ | $0.0_{0.0}$ | $0.0_{0.0}$ |
| Statistical reporting | $12.1_{12.5}$ | $19.6_{10.3}$ | $24.7_{2.8}$ | Mathematics | $5.0_{2.5}$ | $7.1_{2.5}$ | $9.0_{2.0}$ |
| | | | | Medicine | $24.8_{43.2}$ | $46.3_{49.9}$ | $78.2_{41.3}$ |
| | | | | Multidisciplinary | $49.4_{16.7}$ | $73.6_{20.2}$ | $92.3_{14.1}$ |
| | | | | Physics | $0.0_{0.0}$ | $0.0_{0.0}$ | $0.0_{0.0}$ |

Table 17: Mean and standard deviation of pass@$K$ for Gemini-2.0-Flash-Lite-001 ($K \in \{1, 2, 4\}$) by error category (left) and paper category (right). Detailed evaluation results for text-only evaluation of Table 3.

| Error Category | | | | Paper Category | | | |
|---|---|---|---|---|---|---|---|
| Category | pass@1 | pass@2 | pass@4 | Category | pass@1 | pass@2 | pass@4 |
| Data Inconsistency | $2.1_{5.5}$ | $4.0_{7.1}$ | $8.2_{8.3}$ | Biology | $12.9_{21.9}$ | $25.4_{28.6}$ | $49.9_{32.9}$ |
| Equation / proof | $1.1_{1.5}$ | $2.1_{1.7}$ | $3.9_{1.8}$ | Computer Science | $2.1_{3.6}$ | $3.8_{4.2}$ | $6.5_{3.5}$ |
| Experiment setup | $0.0_{0.0}$ | $0.0_{0.0}$ | $0.0_{0.0}$ | Environmental Science | $0.0_{0.0}$ | $0.0_{0.0}$ | $0.0_{0.0}$ |
| Reagent identity | $12.5_{16.1}$ | $25.7_{20.9}$ | $50.1_{24.2}$ | Materials Science | $0.0_{0.0}$ | $0.0_{0.0}$ | $0.0_{0.0}$ |
| Statistical reporting | $3.4_{8.5}$ | $6.5_{10.9}$ | $12.4_{12.5}$ | Mathematics | $0.6_{1.6}$ | $1.1_{2.1}$ | $2.5_{2.5}$ |
| | | | | Medicine | $11.7_{32.2}$ | $26.2_{44.0}$ | $50.7_{50.0}$ |
| | | | | Multidisciplinary | $8.6_{14.6}$ | $16.6_{19.3}$ | $33.0_{22.2}$ |
| | | | | Physics | $0.0_{0.0}$ | $0.0_{0.0}$ | $0.0_{0.0}$ |

Table 18: Mean and standard deviation of pass@$K$ for Claude-3.7-Sonnet:Thinking ($K \in \{1, 2, 4\}$) by error category (left) and paper category (right). Detailed evaluation results for text-only evaluation of Table 3.

| Error Category | | | | Paper Category | | | |
|---|---|---|---|---|---|---|---|
| Category | pass@1 | pass@2 | pass@4 | Category | pass@1 | pass@2 | pass@4 |
| Data Inconsistency | $6.2_{11.6}$ | $12.1_{14.0}$ | $21.1_{13.3}$ | Biology | $0.0_{0.0}$ | $0.0_{0.0}$ | $0.0_{0.0}$ |
| Equation / proof | $4.6_{3.7}$ | $8.4_{4.6}$ | $15.1_{4.9}$ | Computer Science | $7.2_{6.5}$ | $12.4_{7.8}$ | $20.7_{8.9}$ |
| Experiment setup | $0.0_{0.0}$ | $0.0_{0.0}$ | $0.0_{0.0}$ | Environmental Science | $12.4_{21.6}$ | $23.8_{25.0}$ | $39.1_{20.7}$ |
| Reagent identity | $0.0_{0.0}$ | $0.0_{0.0}$ | $0.0_{0.0}$ | Materials Science | $6.2_{16.5}$ | $12.6_{21.7}$ | $24.3_{25.0}$ |
| Statistical reporting | $18.4_{20.6}$ | $31.2_{19.9}$ | $44.3_{11.6}$ | Mathematics | $4.5_{3.9}$ | $8.0_{4.3}$ | $13.6_{3.7}$ |
| | | | | Medicine | $0.0_{0.0}$ | $0.0_{0.0}$ | $0.0_{0.0}$ |
| | | | | Multidisciplinary | $8.0_{14.3}$ | $15.8_{16.7}$ | $26.4_{13.5}$ |
| | | | | Physics | $4.2_{7.2}$ | $7.7_{8.3}$ | $13.0_{6.9}$ |

Table 19: Mean and standard deviation of pass@$K$ for Claude-3.7-Sonnet ($K \in \{1, 2, 4\}$) by error category (left) and paper category (right). Detailed evaluation results for text-only evaluation of Table 3.

| Error Category | | | | Paper Category | | | |
|---|---|---|---|---|---|---|---|
| Category | pass@1 | pass@2 | pass@4 | Category | pass@1 | pass@2 | pass@4 |
| Data Inconsistency | $6.3_{8.1}$ | $11.6_{10.1}$ | $21.5_{10.0}$ | Biology | $6.2_{16.5}$ | $12.0_{21.4}$ | $24.8_{25.0}$ |
| Equation / proof | $4.9_{2.1}$ | $8.9_{3.0}$ | $15.2_{3.4}$ | Computer Science | $6.3_{5.5}$ | $11.8_{6.9}$ | $21.3_{7.0}$ |
| Experiment setup | $0.0_{0.0}$ | $0.0_{0.0}$ | $0.0_{0.0}$ | Environmental Science | $18.4_{24.1}$ | $34.4_{29.3}$ | $63.4_{29.3}$ |
| Reagent identity | $4.1_{11.0}$ | $8.0_{14.2}$ | $16.5_{16.7}$ | Materials Science | $0.0_{0.0}$ | $0.0_{0.0}$ | $0.0_{0.0}$ |
| Statistical reporting | $3.4_{8.5}$ | $6.5_{10.9}$ | $12.4_{12.5}$ | Mathematics | $3.8_{4.2}$ | $6.4_{4.2}$ | $9.9_{3.3}$ |
| | | | | Medicine | $0.0_{0.0}$ | $0.0_{0.0}$ | $0.0_{0.0}$ |
| | | | | Multidisciplinary | $8.9_{14.8}$ | $17.2_{18.9}$ | $33.5_{21.2}$ |
| | | | | Physics | $0.0_{0.0}$ | $0.0_{0.0}$ | $0.0_{0.0}$ |

Table 20: Mean and standard deviation of pass@$K$ for DeepSeek-R1 ($K \in \{1, 2, 4\}$) by error category (left) and paper category (right). Detailed evaluation results for text-only evaluation of Table 3.

| Error Category | | | | Paper Category | | | |
|---|---|---|---|---|---|---|---|
| Category | pass@1 | pass@2 | pass@4 | Category | pass@1 | pass@2 | pass@4 |
| Data Inconsistency | $12.1_{11.7}$ | $20.5_{11.5}$ | $30.2_{6.5}$ | Biology | $7.4_{17.8}$ | $14.3_{22.6}$ | $28.9_{24.7}$ |
| Equation / proof | $16.1_{5.5}$ | $27.8_{6.0}$ | $41.5_{4.3}$ | Computer Science | $13.0_{9.7}$ | $23.8_{10.3}$ | $39.1_{7.7}$ |
| Experiment setup | $0.0_{0.0}$ | $0.0_{0.0}$ | $0.0_{0.0}$ | Environmental Science | $15.7_{23.2}$ | $26.9_{24.9}$ | $41.9_{18.5}$ |
| Reagent identity | $4.9_{11.8}$ | $9.5_{15.1}$ | $19.3_{16.5}$ | Materials Science | $0.0_{0.0}$ | $0.0_{0.0}$ | $0.0_{0.0}$ |
| Statistical reporting | $28.3_{15.7}$ | $43.3_{19.8}$ | $61.3_{18.1}$ | Mathematics | $15.9_{4.1}$ | $26.9_{5.2}$ | $39.7_{5.0}$ |
| | | | | Medicine | $0.0_{0.0}$ | $0.0_{0.0}$ | $0.0_{0.0}$ |
| | | | | Multidisciplinary | $38.4_{20.8}$ | $55.9_{15.6}$ | $65.8_{5.3}$ |
| | | | | Physics | $16.3_{18.0}$ | $28.1_{17.3}$ | $42.2_{9.7}$ |

Table 21: Mean and standard deviation of pass@$K$ for DeepSeek-V3-0324 ($K \in \{1, 2, 4\}$) by error category (left) and paper category (right). Detailed evaluation results for text-only evaluation of Table 3.

| Error Category | | | | Paper Category | | | |
|---|---|---|---|---|---|---|---|
| Category | pass@1 | pass@2 | pass@4 | Category | pass@1 | pass@2 | pass@4 |
| Data Inconsistency | $7.0_{8.2}$ | $12.1_{7.5}$ | $16.2_{2.7}$ | Biology | $7.1_{17.5}$ | $14.2_{22.6}$ | $28.9_{24.7}$ |
| Equation / proof | $1.3_{1.5}$ | $2.6_{2.0}$ | $5.2_{2.1}$ | Computer Science | $1.1_{2.8}$ | $2.2_{3.7}$ | $4.9_{4.1}$ |
| Experiment setup | $0.0_{0.0}$ | $0.0_{0.0}$ | $0.0_{0.0}$ | Environmental Science | $29.3_{24.6}$ | $51.4_{27.0}$ | $76.8_{24.9}$ |
| Reagent identity | $4.7_{11.6}$ | $9.5_{15.1}$ | $19.3_{16.5}$ | Materials Science | $0.0_{0.0}$ | $0.0_{0.0}$ | $0.0_{0.0}$ |
| Statistical reporting | $0.0_{0.0}$ | $0.0_{0.0}$ | $0.0_{0.0}$ | Mathematics | $0.7_{1.7}$ | $1.4_{2.3}$ | $2.9_{2.5}$ |
| | | | | Medicine | $0.0_{0.0}$ | $0.0_{0.0}$ | $0.0_{0.0}$ |
| | | | | Multidisciplinary | $0.0_{0.0}$ | $0.0_{0.0}$ | $0.0_{0.0}$ |
| | | | | Physics | $0.0_{0.0}$ | $0.0_{0.0}$ | $0.0_{0.0}$ |

Table 22: Mean and standard deviation of pass@$K$ for Qwen3-235A-22B ($K \in \{1, 2, 4\}$) by error category (left) and paper category (right). Detailed evaluation results for text-only evaluation of Table 3.

| Error Category | | | | Paper Category | | | |
|---|---|---|---|---|---|---|---|
| Category | pass@1 | pass@2 | pass@4 | Category | pass@1 | pass@2 | pass@4 |
| Data Inconsistency | $8.4_{8.3}$ | $15.8_{9.5}$ | $26.7_{8.2}$ | Biology | $0.0_{0.0}$ | $0.0_{0.0}$ | $0.0_{0.0}$ |
| Equation / proof | $17.1_{6.9}$ | $28.1_{6.1}$ | $40.7_{3.3}$ | Computer Science | $17.9_{5.7}$ | $29.9_{7.4}$ | $43.2_{5.6}$ |
| Experiment setup | $0.0_{0.0}$ | $0.0_{0.0}$ | $0.0_{0.0}$ | Environmental Science | $7.6_{18.0}$ | $16.0_{23.3}$ | $33.2_{23.6}$ |
| Reagent identity | $5.7_{12.5}$ | $11.6_{15.9}$ | $22.3_{15.7}$ | Materials Science | $0.0_{0.0}$ | $0.0_{0.0}$ | $0.0_{0.0}$ |
| Statistical reporting | $25.9_{20.6}$ | $44.4_{18.9}$ | $64.8_{12.3}$ | Mathematics | $12.7_{6.3}$ | $20.6_{6.3}$ | $31.2_{4.9}$ |
| | | | | Medicine | $17.0_{37.6}$ | $34.8_{47.7}$ | $66.9_{47.1}$ |
| | | | | Multidisciplinary | $45.2_{24.4}$ | $74.6_{25.0}$ | $98.2_{7.5}$ |
| | | | | Physics | $16.6_{9.7}$ | $28.7_{12.8}$ | $43.2_{10.2}$ |

Table 23: Mean and standard deviation of pass@$K$ for Qwen2.5-VL-72B-Instruct ($K \in \{1, 2, 4\}$) by error category (left) and paper category (right). Detailed evaluation results for text-only evaluation of Table 3.

| Error Category | | | | Paper Category | | | |
|---|---|---|---|---|---|---|---|
| Category | pass@1 | pass@2 | pass@4 | Category | pass@1 | pass@2 | pass@4 |
| Data Inconsistency | $18.9_{10.4}$ | $31.0_{9.1}$ | $42.2_{8.3}$ | Biology | $23.2_{38.1}$ | $41.6_{43.3}$ | $70.8_{36.6}$ |
| Equation / proof | $0.4_{1.0}$ | $0.8_{1.3}$ | $1.8_{1.5}$ | Computer Science | $0.0_{0.0}$ | $0.0_{0.0}$ | $0.0_{0.0}$ |
| Experiment setup | $0.0_{0.0}$ | $0.0_{0.0}$ | $0.0_{0.0}$ | Environmental Science | $34.4_{34.5}$ | $55.4_{31.7}$ | $79.6_{24.6}$ |
| Reagent identity | $15.5_{25.4}$ | $27.7_{28.8}$ | $47.2_{24.4}$ | Materials Science | $8.2_{18.6}$ | $15.2_{23.0}$ | $28.1_{24.8}$ |
| Statistical reporting | $6.9_{11.2}$ | $13.3_{12.5}$ | $21.2_{9.0}$ | Mathematics | $0.0_{0.0}$ | $0.0_{0.0}$ | $0.0_{0.0}$ |
| | | | | Medicine | $0.0_{0.0}$ | $0.0_{0.0}$ | $0.0_{0.0}$ |
| | | | | Multidisciplinary | $22.9_{15.5}$ | $41.3_{17.9}$ | $60.7_{12.8}$ |
| | | | | Physics | $0.0_{0.0}$ | $0.0_{0.0}$ | $0.0_{0.0}$ |

Table 24: Mean and standard deviation of pass@$K$ for Qwen2.5-VL-32B-Instruct ($K \in \{1, 2, 4\}$) by error category (left) and paper category (right). Detailed evaluation results for text-only evaluation of Table 3.

| Error Category | | | | Paper Category | | | |
|---|---|---|---|---|---|---|---|
| Category | pass@1 | pass@2 | pass@4 | Category | pass@1 | pass@2 | pass@4 |
| Data Inconsistency | $4.6_{7.4}$ | $8.9_{8.3}$ | $14.1_{6.0}$ | Biology | $0.0_{0.0}$ | $0.0_{0.0}$ | $0.0_{0.0}$ |
| Equation / proof | $0.0_{0.0}$ | $0.0_{0.0}$ | $0.0_{0.0}$ | Computer Science | $0.0_{0.0}$ | $0.0_{0.0}$ | $0.0_{0.0}$ |
| Experiment setup | $0.0_{0.0}$ | $0.0_{0.0}$ | $0.0_{0.0}$ | Environmental Science | $0.0_{0.0}$ | $0.0_{0.0}$ | $0.0_{0.0}$ |
| Reagent identity | $4.7_{11.6}$ | $9.5_{15.1}$ | $19.3_{16.5}$ | Materials Science | $0.0_{0.0}$ | $0.0_{0.0}$ | $0.0_{0.0}$ |
| Statistical reporting | $0.0_{0.0}$ | $0.0_{0.0}$ | $0.0_{0.0}$ | Mathematics | $0.0_{0.0}$ | $0.0_{0.0}$ | $0.0_{0.0}$ |
| | | | | Medicine | $14.2_{34.9}$ | $28.5_{45.2}$ | $57.8_{49.4}$ |
| | | | | Multidisciplinary | $9.1_{14.9}$ | $17.7_{16.6}$ | $28.3_{11.9}$ |
| | | | | Physics | $0.0_{0.0}$ | $0.0_{0.0}$ | $0.0_{0.0}$ |

Table 25: Mean and standard deviation of pass@$K$ for Llama-4-Maverick ($K \in \{1, 2, 4\}$) by error category (left) and paper category (right). Detailed evaluation results for text-only evaluation of Table 3.

| Error Category | | | | Paper Category | | | |
|---|---|---|---|---|---|---|---|
| Category | pass@1 | pass@2 | pass@4 | Category | pass@1 | pass@2 | pass@4 |
| Data Inconsistency | $0.0_{0.0}$ | $0.0_{0.0}$ | $0.0_{0.0}$ | Biology | $6.7_{17.0}$ | $13.9_{22.4}$ | $28.3_{24.8}$ |
| Equation / proof | $0.8_{1.3}$ | $1.7_{1.8}$ | $3.5_{2.0}$ | Computer Science | $1.1_{2.8}$ | $2.2_{3.7}$ | $4.9_{4.1}$ |
| Experiment setup | $0.0_{0.0}$ | $0.0_{0.0}$ | $0.0_{0.0}$ | Environmental Science | $0.0_{0.0}$ | $0.0_{0.0}$ | $0.0_{0.0}$ |
| Reagent identity | $4.5_{11.4}$ | $9.3_{14.9}$ | $18.9_{16.5}$ | Materials Science | $0.0_{0.0}$ | $0.0_{0.0}$ | $0.0_{0.0}$ |
| Statistical reporting | $0.0_{0.0}$ | $0.0_{0.0}$ | $0.0_{0.0}$ | Mathematics | $0.0_{0.0}$ | $0.0_{0.0}$ | $0.0_{0.0}$ |
| | | | | Medicine | $0.0_{0.0}$ | $0.0_{0.0}$ | $0.0_{0.0}$ |
| | | | | Multidisciplinary | $4.7_{11.6}$ | $9.7_{15.2}$ | $18.6_{16.6}$ |
| | | | | Physics | $0.0_{0.0}$ | $0.0_{0.0}$ | $0.0_{0.0}$ |

Table 26: Mean and standard deviation of pass@$K$ for Llama-4-Scout ($K \in \{1, 2, 4\}$) by error category (left) and paper category (right). Detailed evaluation results for text-only evaluation of Table 3.

| Error Category | | | | Paper Category | | | |
|---|---|---|---|---|---|---|---|
| Category | pass@1 | pass@2 | pass@4 | Category | pass@1 | pass@2 | pass@4 |
| Data Inconsistency | $6.9_{8.2}$ | $13.1_{9.7}$ | $23.9_{9.2}$ | Biology | $6.5_{16.8}$ | $13.1_{22.0}$ | $29.6_{24.6}$ |
| Equation / proof | $0.8_{1.3}$ | $1.5_{1.5}$ | $2.6_{1.0}$ | Computer Science | $2.3_{3.7}$ | $4.1_{4.2}$ | $7.2_{2.8}$ |
| Experiment setup | $0.0_{0.0}$ | $0.0_{0.0}$ | $0.0_{0.0}$ | Environmental Science | $14.1_{22.5}$ | $26.2_{25.0}$ | $42.2_{18.2}$ |
| Reagent identity | $4.3_{11.2}$ | $8.7_{14.6}$ | $19.7_{16.4}$ | Materials Science | $6.5_{16.8}$ | $13.1_{22.0}$ | $29.6_{24.6}$ |
| Statistical reporting | $0.0_{0.0}$ | $0.0_{0.0}$ | $0.0_{0.0}$ | Mathematics | $0.0_{0.0}$ | $0.0_{0.0}$ | $0.0_{0.0}$ |
| | | | | Medicine | $0.0_{0.0}$ | $0.0_{0.0}$ | $0.0_{0.0}$ |
| | | | | Multidisciplinary | $0.0_{0.0}$ | $0.0_{0.0}$ | $0.0_{0.0}$ |
| | | | | Physics | $0.0_{0.0}$ | $0.0_{0.0}$ | $0.0_{0.0}$ |

