# OpenReview forum: "When AI Co‑Scientists Fail: SPOT—a Benchmark for Automated Verification of Scientific Research"
_ICLR.cc/2026/Conference — Submitted to ICLR 2026_

### Official Review · Reviewer_KTDn · 2025-10-19

**Soundness:** 4
**Presentation:** 3
**Contribution:** 3
**Rating:** 6
**Confidence:** 3

**Summary:**

The paper introduces **SPOT**, a benchmark designed to evaluate large language models (LLMs) on the task of **automated scientific error detection** across full-length multimodal scientific manuscripts. Unlike previous work focused on factual claim verification or peer-review generation, SPOT targets the *backward pass* of science—verification rather than generation.

The dataset includes **83 papers with 91 validated errors**, spanning 10 scientific domains and 6 types of errors (e.g., equation/proof mistakes, figure duplications, data inconsistencies). Importantly, each error is *confirmed by the original authors* or appears in published errata/retractions.

State-of-the-art models (OpenAI o3, GPT-4.1, Gemini 2.5, Claude 3.7, Llama 4, etc.) are evaluated. Results show **very low recall and precision** (best recall ≈ 21.1%, precision ≈ 6.1%), highlighting a substantial gap between current LLM capabilities and reliable scientific verification. Models often show low confidence, poor reproducibility across runs, and frequent hallucinations.

Two expert-led case studies (in mathematics and materials science) reveal that models either miss core errors or hallucinate incorrect issues, confirming their unreliability in this domain.

**Strengths:**

* **Novelty and Relevance**: The paper addresses a crucial but underexplored problem—scientific verification rather than generation—highly relevant to the development of AI Co-Scientists.
* **High-Quality Benchmark Construction**:
  * Errors are cross-validated by human experts and confirmed by original authors.
  * Covers multimodal inputs (text + figures/tables), long contexts (~12k tokens), and multiple scientific domains.
- **Meaningful Empirical Findings**:
  * Demonstrates that models like o3 achieve only 21% recall and 6% precision.
  * Shows a lack of reproducibility across runs and low confidence calibration.

**Weaknesses:**

* **Author-Confirmed Errors Bias**:
  * Only includes errors acknowledged by authors, which may omit subtle or controversial errors.
  * Treats unannotated model-detected errors as false positives by default, even if they might be valid.
* **Naive Baselines**:
  * As there are numerous studies for LLM-based reviewing, the authors could adopt them to this setup. However, they only tested a simple prompting-based approach.

**Questions:**

- Line 052: Another related work: [*Task Contamination: Language Models May Not Be Few-Shot Anymore*](https://ojs.aaai.org/index.php/AAAI/article/view/29808)
- Some potential related work:
  - [*ReviewScore: Misinformed Peer Review Detection with Large Language Models*](https://arxiv.org/abs/2509.21679)
  * [*Position Paper: How Should We Responsibly Adopt LLMs in the Peer Review Process?*](https://openreview.net/forum?id=KZ3NspcpLN)

---

### Official Review · Reviewer_mXz8 · 2025-10-30

**Soundness:** 2
**Presentation:** 3
**Contribution:** 3
**Rating:** 2
**Confidence:** 3

**Summary:**

The paper has several genuine positives: it identifies a real and timely gap in moving AI “co-scientists” from generation to verification, and instantiates that gap in a realistic, hard, long-context, multimodal benchmark built from author-confirmed errors. It also evaluates a broad slate of current frontier and open models and shows a compelling negative result, which is useful to the community. However, for a top-tier venue, there are multiple blocking weaknesses that could be hard to fix in one revision: the benchmark is small and somewhat source-biased; the evaluation protocol assumes exhaustiveness and therefore penalizes models for discovering genuinely new errors; the scoring pipeline itself uses an unvalidated LLM-based matcher; and no human-expert baseline or statistical significance is reporting to anchor the surprisingly low numbers.

**Strengths:**

S1: I like the clear framing of the problem. The experiment setting is somewhat realistic: long-context, multimodal, full paper input (~12k tokens + ~18 images) that better matches actual scholarly reading tasks.

S2: The dataset contributed is good. High-credibility ground truth via author confirmation and human sanity checks, so the target errors are noncontroversial and actually present in the PDFs.

S3: Insightful qualitative case studies that show how and where current models fail (e.g., student-like mistakes, wrong-section focus).

S4: Broad, modern evaluation across 10 prominent proprietary and open MLLMs, with pass@K and multi-run protocol, yielding a compelling negative result.

**Weaknesses:**

W1: Dataset scale is modest (83 manuscripts, 91 errors), so the very strong negative result rests on a relatively small sample and may not cover the full diversity of real scholarly errors.

W2: Although the evaluation protocol is clearly specified, it assumes the benchmark is exhaustive; genuinely new, model-found errors are still counted as false positives, which likely underestimates the practical usefulness of stronger detectors.

W3: Although outputs are scored in a reproducible way, the scoring depends on an LLM-based matcher whose own accuracy/robustness is not reported, leaving an unmeasured source of evaluation noise.

W4: No human-expert / domain-PhD baseline is reported, so we cannot tell whether the observed 6–21% performance reflects model weakness or task intrinsic difficulty.

W5: Even though many strong models are compared, the paper does not report statistical significance or variance across the 8-run setup, making it hard to judge ranking stability.

W6: Multimodality is central to the task, but the paper does not foreground a clear, in-main-text ablation on text-only vs multimodal and per-error-type performance, leaving open whether figures are currently too noisy.

W7: Only single-prompt, single-agent model runs are considered; there are no tool-augmented, retrieval-enhanced, or multi-agent baselines that could narrow the reported gap.

**Questions:**

Q1: Could you provide a per-domain / per-error-type breakdown (e.g. math/equations vs bio/figures vs CS/protocol) to show that the negative results are not an artifact of one or two overrepresented categories, and to demonstrate that 83/91 is still informative across domains?

Q2: Since the current set comes only from WITHDRARXIV/PubPeer and author-confirmed items, can you add a small “non-public” or “non-author-confirmed” slice (even 10–15 items curated by domain experts) to test whether the models fail similarly on errors that have not already surfaced publicly?

---

### Official Review · Reviewer_iBoP · 2025-10-31

**Soundness:** 2
**Presentation:** 3
**Contribution:** 2
**Rating:** 2
**Confidence:** 3

**Summary:**

1. Contributes SPOT and describes the collection/construction process of a benchmark consisting of 83 published papers with 91 error instances that prompt errata or retractions, validated using actual authors and human annotators.
2. Analysis of model confidence and evaluation of current SOTA agents, all of which have low performance, showing that this is a challenging problem.

**Strengths:**

1. SPOT’s pipeline (real-world Withdrawn ArXiv + PubPeer → human + author checks, training-cutoff aware) yields a high-fidelity dataset of genuine scientific errors, improving ecological validity over synthetic bugs.
2. Clear positioning vs prior work and reasonable filtering criteria. The paper also reads cleanly and makes it very easy to see how this dataset fills a gap (especially with recent workshops like Agents4Science).
3. Broad SOTA agent sweep with confidence calibration, breakdowns, and illustrative case studies convincingly shows the task is hard (low performance across the board). In addition to the headline result that no model surpasses 21.1% recall or 6.1% precision (Table 2, p. 5), the paper probes calibration (Fig. 4), multi-modality vs. text-only (Table 3), long-context effects, and test-time scaling; expert case studies diagnose concrete failure modes and even surface a genuine unannotated units error (10,000×) in materials science.

**Weaknesses:**

1. Limited size of dataset, task instances, and size of each sub category

1a. 83 papers / 91 errors with very small per-category counts make it hard to interpret statistical significance of some results and overall generalizability, especially for subjects/categories with very low paper/error counts.

1b. For example, engineering has less than 5 errors (which may not be representative of a real-world set of engineering paper errors). Thus, in my opinion, the title of “automatic verification of scientific research” is somewhat over-general given this.



2. The dataset pipeline clearly selects for presence of true positive errors occurrences (only selecting papers where there was an error and a subsequent retractions), but the metrics you report show a high degree of false-positives (low-precision).

2a. Is this due to LMs actually identifying false errors, or could there be additional errors in manuscripts that escaped the review process and thus are contributing to this high false-positive rate?



3. Metric choices and difficulty in measuring improvement on this benchmark:

3a. I think not having a single primary metric makes it harder to also see the usefulness of this benchmark.

3b. From examining the pass@K formulation (Equation 2), if a model reports the entire paper as errors, I would have a 100% pass@K (i.e. this metric does not penalize false positives). In other words, this specific metric answers “is the true positive somewhere in my predicted set”?

As an aside, I’m also not sure this matches usage of pass@K in code/agent/free-response question benchmarks. In other words, is an instance “solved” if I flag the whole paper as an error?

3c. Similarly, recall as a metric here suffers the same thing: predicting the entire paper as an error would also capture all true positives, report no false negatives, and have a recall of 100%. Precision as a metric seems to make sense here, but hill-climbing on this metric alone means we just are calibrating models to not falsely report true-positives: the question then arises if this is actually measuring improvement on “automatic validation of scientific research”

3d. In terms of alternative metrics I’d suggest, in actuality, this seems to be a segmentation problem (i.e. segment which parts of the paper have errors) and metrics like intersection over union (IoU) used in computer vision or bounding box tasks seem like a much better choice here and unify all your metrics into one.

**Questions:**

1. Unclear if model failures are from needle-in-haystack difficulty or missing domain knowledge:

1a. Are there statistics on size of error regions in papers (i.e relative to whole paper size, how big is the average error)? Curious on if errors are difficult to find because they’re very tiny (needle in haystack) or if it’s due to lack of knowledge/difficulty reasoning over certain topics?

3. What does pass@k mean here? Does this mean cases where the set of predicted LM errors exactly matches the true set of flagged errors (or just that the true positives match)?

4. Nit: What do the subscripts in Table 2 mean (error bars from multiple runs)?

---

### Official Review · Reviewer_tZF1 · 2025-11-01

**Soundness:** 2
**Presentation:** 2
**Contribution:** 2
**Rating:** 4
**Confidence:** 3

**Summary:**

This paper introduces SPOT, a multimodal benchmark for evaluating large language models as scientific verifiers rather than generators. SPOT includes 83 real papers with 91 confirmed errors across ten domains, validated by original authors and human experts. Evaluations of ten leading LLMs show very low precision and recall, revealing that current models fail at reliable academic error detection. Expert case studies further show student-level misunderstandings and poor calibration.

**Strengths:**

* The paper shifts focus from LLMs as generators to verifiers of scientific work, addressing a crucial yet underexplored aspect of AI-assisted research. The proposed new task is interesting to delve into.


* SPOT is carefully curated with author-acknowledged and human-validated errors across 10 domains, ensuring strong data credibility and realism, also well addressed the ethical considerations.


* The benchmark includes full scientific papers (text + figures), testing models under authentic, long-context, multimodal conditions rarely covered in prior work.


* Evaluations across 10 state-of-the-art proprietary and open LLMs with consistent metrics (precision, recall, pass@K) and multiple runs offer robust, reproducible insights.


* Expert-led case studies expose systematic reasoning failures and calibration issues in current models, providing valuable diagnostic evidence for future research.

**Weaknesses:**

* The dataset is relatively small (83 papers, 91 errors), limiting the generalizability of the findings.

* While SPOT presents an ambitious attempt to benchmark AI systems for scientific verification, the task definition itself is arguably ill-posed. The benchmark assumes that a language model, given only the text and figures of a paper, can identify deep scientific or methodological errors—something that often requires domain expertise, experimental data, or external validation beyond the document.

* The study lacks a human baseline, making it difficult to contextualize the reported precision and recall numbers. Also the results on open-source models are quite bad score, which rasing the concern of the task validation. I would suggest to decompose the task by the difficulty, and solve the problem step-by-step, rather than direct prompt with whole paper.

* Evaluation is binary and does not capture partial correctness or the quality of explanations provided by models.

* Although SPOT is multimodal, the analysis of cross-modal reasoning and figure understanding remains superficial.

* The work does not clearly define or formalize what constitutes “scientific verification ability,” relying solely on empirical evidence.

* Paper writing is still limited. For example, some common sense equations (Precision, recall, pass@K) should be removed. The input and output of example is not clearly explained (line 195-197 and appendix F). Also F.2 exceeds the page layout.

**Questions:**

1. How do the authors plan to address the limited dataset scale in future work — for example, by semi-automated error mining or expanding to other disciplines beyond the current 83 papers?

2. Since all non-annotated detections are counted as false positives, did the authors manually check whether some of these might correspond to real but unrecorded errors?

3. Could the authors provide a comparison with human reviewers (e.g., graduate students or peer reviewers) to contextualize model performance and better interpret the low precision/recall numbers?

4. Direct prompting is not an effective way to conduct such challenging task, why not try multi-agent framework?

---

### Meta-Review · Area_Chair_P9RX · 2026-01-07

**Summary:**

Strengths

- The benchmark is extremely realistic (full scientific papers, long context, multimodal; iBoP, tZF1, mXz8, KTDn)
- Clearly written paper; obvious gap was filled (iBoP, KTDn)
- Good error analysis (mXz8, tZF1)

Weaknesses

- Small dataset (tZF1, iBoP, mXz8)
- Lacks a human baseline (tZF1, mXz8)
- Binary evaluation (tZF1)
- Ill-defined metrics (iBoP)
- Lacking more recent LLM developments (tool use, reasoning, multiple turns) (mXz8, KTDn)

**Reviewer Concerns:**

N/A, no rebuttal. All concerns are still outstanding.

**Reviewer Scores:**

N/A, no rebuttal

---

### Decision · Program_Chairs · 2026-01-26

Reject